# Are the Time-Poor Willing to Pay More for Online Grocery Services? When 'No' Means 'Yes'

**Ellen Van Droogenbroeck** * and **Leo Van Hove**

Department of Applied Economics, Vrije Universiteit Brussel (Free University of Brussels),
1050 Brussels, Belgium; Leo.Van.Hove@vub.be
*   Correspondence: Ellen.Van.Droogenbroeck@vub.be

**Abstract:** This paper investigates consumers' willingness to pay (WTP) for click-and-collect grocery services. In particular, we analyze whether the time-pressed are willing to pay higher fees. We exploit a survey among 572 customers of two Belgian supermarket chains—both users and non-users. We test our model for three (increasingly narrow) samples: all respondents, respondents with a non-zero WTP, and current users. Our key finding relates to the latter sample. Surprisingly, if we use the WTP measure put forward in the literature, the answer to our research question is 'no': we find no significant relationship between users' perceived time pressure and the maximum service cost per order they are willing to pay. However, on closer scrutiny this 'no' in fact means 'yes': our finding implies that in the face of increasing fees the time-pressed are willing to maintain their current, higher order frequency for as long as the other users. The maximum *total* cost they are willing to incur over a given period is thus higher. This said, the absence of a relationship between time pressure and the WTP per order does limit the opportunities for e-grocers to price discriminate, as is suggested in the literature. A further complication is that we find no clear pattern between perceived time pressure and the use of specific time slots.

**Keywords:** e-grocery services; willingness to pay; perceived time pressure; price discrimination

## 1. Introduction

Online grocery shopping has instigated the return of traditional, pre-supermarket delivery concepts [1,2]. Before the advent of the supermarket, grocers drove around with vans to deliver the goods to the customer's home. Subsequently, supermarkets outsourced the delivery to the customer—together with the picking and bagging. Online grocery shopping brought the picking, packing, and—at least in the home delivery model—also the delivery back to the retailer. As a result, retailers face additional logistics costs [3].

To cover these extra costs, e-grocers typically charge a service fee. The literature argues that users willingly pay these fees because they think of grocery shopping as a chore and appreciate the convenience of the online service. According to this 'convenience hypothesis' [4], cost recovery should thus not be a problem. (Kotzab and Teller [4] formulate the 'convenience hypothesis' for the home delivery model; however, supermarkets with a click-and-collect service also incur additional costs.)

However, the reality is less straightforward. In the early days of e-commerce, several start-ups, such as Webvan, failed. Even today e-grocers struggle with their costs. In Germany, leading grocery chain Kaufland launched its online channel in 2016, only to close it one year later. The reason was concerns about the potential for future profitability [5]. Overall, consumer adoption of e-grocery shopping is effectively still rather low. In the European Union 27, only 19% of all consumers (i.e., people between the age of 16 and 74) bought groceries online in 2020 [6]. (Note that most countries collected data in the first half of 2020. As a consequence, the impact of the COVID-19 pandemic is not (entirely) reflected.)

These observations highlight the need for research into the service costs charged by retailers, and, most importantly, into consumers' willingness to pay such fees. In the literature, several authors suggest that e-grocers should price discriminate across time slots and days. Charging higher fees for popular and/or rush-hour time slots would, so the argument goes, enable supermarkets to steer demand, create more cost-effective schedules, and improve profitability [1,3,7,8]. However, most studies take consumers' willingness to pay (WTP) for granted, even though research on the subject is scant, as a recent systematic review of e-grocery studies shows [9].

Moreover, the majority of the existing studies [4,8,10,11] examine home delivery services rather than the click-and-collect model (where customers collect their groceries at a pick-up-point). To the best of our knowledge, only Seitz et al. [12] and Milioti et al. [13] have investigated the WTP for click-and-collect services. This lack of attention is surprising because the click-and-collect model is dominant in countries such as France and Belgium [14–16] and strongly on the rise in important markets such as the US and the UK [17,18].

The goal of the present paper is therefore to investigate consumers' WTP for this type of service. In particular, we analyze whether time-pressed consumers are willing to pay higher fees. To that end, we exploit a self-conducted survey among 572 customers of two Belgian supermarkets—both users (105) and non-users (467) of online grocery services—and test the proposed relationships by means of a system of regression equations.

In doing so, our paper contributes to the broader literature on the last-mile issue—the delivery to the customer's home. There are several studies that evaluate the financial and environmental performance of different last-mile distribution concepts; see, among others, Belavina et al. [19], Fikar et al. [20], and Hübner et al. [3]. In a recent study for Germany, Wollenburg et al. [21], for example, examine how delivery modes—offered in different combinations of delivery cost and velocity—can be used to steer customers across channels. However, their study focuses on fashion omni-channel retailing, and the authors distinguish only between delivery free of charge or at a cost. Our paper adds insights to this literature concerning the level of, and variation in, consumers' WTP—at least for one type of product and for a specific delivery method. In so doing, our paper also exposes the opportunities, if any, for e-grocers to price discriminate.

The remainder of the paper is structured as follows. After a discussion of the literature, we develop our conceptual model and list the hypotheses. Next, we explain the methodology as well as the data collection process. Subsequently, we examine the composition of the sample and present the results of the regression analyses. Importantly, we test our model for three (increasingly narrow) samples: all respondents, respondents with a non-zero WTP, and current users of e-grocery services. We end the paper with an in-depth discussion of the most important findings together with a conclusion.

## 2. Online Grocery Services: Theoretical Background

### 2.1. The Logistics

The retail food industry in general and the grocery channel in particular have undergone significant changes throughout the past centuries [22]. In continuous attempts to respond to changes in the consumer population and the advancement of technology, the industry has been evolving from independent and unaffiliated corner stores (until the mid-1800s), over chain grocery retailing (in the beginning of the twentieth century), to self-service chain stores and the opening of the first modern supermarkets in the US (in 1930); for a more complete overview, see Stanton [22]. The supermarket concept then gradually spread around the world and the following decades were characterized by several major developments, such as the introduction of hypermarkets, (hard) discounters, club stores, convenience stores, and private labeling [22]. In the late nineties of the previous century, the (food) retail sector was again disrupted, now by the advent of the Internet and the arrival of e-commerce, allowing companies to sell their goods and services via the world wide web.

Developments in e-grocery have had a significant impact on food supply chains, and three categories of e-grocers have emerged [23,24]. For one, many bricks-and-mortar supermarkets opened an online channel next to their traditional offline channel, thus becoming so-called bricks-and-clicks. This is the category that we focus on in the present paper. Second, several online-only supermarkets (without physical stores), such as Ocado in the UK and Peapod in the US, now sell groceries online and fulfill deliveries directly from their warehouses. Finally, in recent years, large Internet-based retailers, such as Amazon, have started harnessing their e-commerce expertise to build their own online grocery shops.

In the process of online grocery shopping, the (weekly) visit to a physical supermarket has been replaced by ordering from home (or any other place) electronically—via the store's website or app [25]. The retailer then takes care of the picking and packing of the goods, either in-store, at separate fulfilment centers, or at central warehouses [3]. Where the delivery is concerned, the idiosyncratic features of (particular) groceries, including perishability, sensitivity to temperature, shelf life, and relatively low item value, make the 'last mile' a major challenge [24,26]. Overall, there are two commonly applied approaches: home delivery and click-and-collect. For a detailed overview of all last-mile delivery practices in omnichannel, we refer to the work of Hübner et al. [3] and Risberg and Jafari [27].

For home delivery, there are two major options [3]. The most popular is attended home delivery. This implies that the retailer delivers the groceries, within a certain timeframe, to an agreed point of reception (the customer's home or workplace), where the customer accepts the order. Although this option is convenient, it also creates complexity as the customer is under constraint to wait for the order to be delivered [3,24]. E-grocers try to mitigate this drawback by offering narrow, one-hour delivery windows, but this, in turn, leads to complex vehicle routing problems and high fulfillment costs. Moreover, it is difficult to pass on these costs to the customers, given their high sensitivity to delivery charges. As a result, attended home delivery is characterized by competitive pressure over the quality of the service [24,28].

This said, pensioners, students, or people working from home may be more flexible, in that they might be willing to accept uncertainty over the exact time slot in return for an incentive. The latter does not necessarily need to take the form of a lower delivery charge, but can also be non-monetary, by emphasizing the environmental benefits of more flexible delivery options. Several retailers, including Albert Heijn in the Netherlands (and Belgium) and Tesco in the UK, exploit this option by offering so-called 'Flexi Saver Slots' next to their regular one-hour slots. That is, customers select a four-hour window and on the day of delivery they receive a notification of the one-hour slot during which the groceries will be delivered. Taking this flexibility to a higher level, Strauss et al. [28] propose "customizable flexible slots", allowing customers to tailor the slots to their own schedule by selecting a series of available, not necessarily adjacent slots for a fixed discount. For example, "pick any three standard one-hour slots for a delivery charge of only £X" [28] (p. 1023).

Despite all these attempts to promote flexibility in the selection of time slots, the failure rate in attended home delivery is still sizeable. Customers often are absent, be it due to their own fault (failing to remember, emergency at work or with the children, etc.) or due to delayed delivery because of traffic [24]. The unattended option is considerably more convenient for both retailer and consumer, as the latter does not need to be present to accept the order. Instead, the groceries are delivered to a (cooled) reception box of the household, a shared delivery box near the customer's home, or even in-car or in-fridge [3,26]. Downsides include the investment in reception facilities and concerns about safe and secure delivery [3,26].

In the click-and-collect model, for its part, it is the customer who does the picking-up within a convenient time frame. This can take place at the supermarket itself (either in-store or at an attached pick-up point), at a stand-alone pick-up point, or at a drive-through station attached to the store or at another location [3,14]. Although this approach, thus, still requires a trip to the store or the pick-up point, the click-and-collect system is convenient and flexible, as collection is rapid and can often be fitted into consumers' existing trips and schedules.

Although pick-up is the main channel in countries with a high store density, such as France and Belgium, attended home delivery is still generally the predominant model [3]. Yet, recent data for the US from analytics and strategic insight firm Brick Meets Click show that "if you are a retailer that is in delivery only, there is a sizeable segment of your market that is not attracted to your service" [29]. As traditional bricks-and-clicks grocers have to compete with large Internet-based retailers such as Amazon, which offer home delivery within 30 min to one hour, (further) developing click-and-collect can be a means to maintain market share and/or conquer new territories [14]. The deployment of click-and-collect has been accelerated by the COVID-19 crisis. During the pandemic, several grocers converted stores to pick-up locations out of necessity. However, online grocery demand continues to stay high [29–31]. As a result, retailers no longer wonder whether they need click-and-collect; their main issue is now which specific solution is the most efficient and attractive [29]. In short, given its increased popularity, academic research on the currently underexplored domain of click-and-collect becomes crucial [14,16].

### 2.2. How Convenient Is E-Grocery Shopping?

Buying groceries is a necessary task that consumers often perceive as a constraint. Grocery shopping is associated with efficiency rather than enjoyment. It is thus not surprising that the main reasons for ordering groceries online relate to convenience, and more in particular time saving [32–34].

In the marketing literature, convenience was initially used as an adjective describing a class of consumer goods, namely, "those [goods] customarily purchased at easily accessible stores" [35] (p. 282). Over time, convenience became a concept on its own, with an emphasis on the reduction in two key non-monetary costs: time and effort [36–38]. Researchers started describing convenience as a multidimensional or second-order construct and they distinguished between two (closely related) types. Seiders et al. [39] presented *retail convenience* as time and effort costs associated with shopping in a retail environment. Berry et al. [40] (p. 1) introduced *service convenience*, defined as "a consumer's time and effort perceptions related to buying or using a service". More in particular, the time and effort characteristics of convenience were categorized into activity-based dimensions, which mirror the activities consumers perform to purchase a product or a service. Whereas early studies were restricted to the conventional brick-and-mortar retailing environment [36,38–41], more recent research explores the differences in convenience between shopping in-store and online buying [42], or focuses on the convenience dimensions that are typical for the e-commerce setting [43]. Jiang et al. [43], in their study for Hong Kong, identify five key dimensions of online shopping convenience: access, search, evaluation, transaction, and possession (post-purchase) convenience. If one applies this framework to the act of shopping for groceries online, as we do in Table 1, it is clear that the entire process (from preparing shopping lists to storing the goods) is associated with substantial levels of convenience in each of the five dimensions.

**Table 1.** Dimensions of convenience applied to e-grocery services.

| Dimension | Definition | Applied to Online Grocery Shopping |
|---|---|---|
| Access convenience | The speed and ease with which consumers can reach a retailer. | - Ordering can be made anytime (24 h a day, 7 days a week), anywhere (at home, at work, . . . ) and from any device;<br>- Time flexibility: the ordering process can be fragmented (e.g., one can start the order at work during lunch, continue in the train back home and confirm it in the evening at home);<br>- Avoidance of crowds, reduced waiting time, less time/effort spent on traveling to physical stores and parking (the latter mainly relates to home delivery services). |

**Table 1.** *Cont.*

| Dimension | Definition | Applied to Online Grocery Shopping |
|---|---|---|
| Search convenience | The speed and ease with which consumers can identify and select products they wish to buy. | - E-grocer websites are easy to navigate, include clear product classifications, search functions and filters;<br>- Multiple shopping lists can easily be composed and saved;<br>- Several shopping lists (last order, recent purchases, most bought items, . . . ) are automatically provided;<br>- Products can be added by scanning the barcode with the retailer's app. |
| Evaluation convenience | The speed and ease with which consumers can access detailed yet easy-to-understand product descriptions. | - Product descriptions are accompanied by pictures;<br>- Apart from the basic information (product description, price, size, . . . ), retailers often provide additional details on nutritional values and allergens, origin, claims/labels (vegan, Fairtrade, sustainable seafood, . . . ), and producer information;<br>- Special offers and discounts are clearly indicated;<br>- Preferences or concerns can be specified in the order (for example: bananas not too green). |
| Transaction convenience | The speed and ease with which consumers can effect or amend transactions. | - The order can be placed through the website or the app;<br>- Order confirmation is provided, including detailed information on discounts and service/delivery costs;<br>- Simple and convenient payment methods are in place: either online or when the groceries are collected/delivered to the door;<br>- Reservations can be altered (both basket and delivery time) until one day or x hours before collection/delivery;<br>- Customer service can be contacted via website and/or telephone. |
| Possession (post-purchase) convenience | The speed and ease with which consumers can obtain desired products. | - (One-hour) time slots can be chosen by the customer. Most retailers offer next-day delivery but some also provide same-day delivery;<br>- The shopping activity is limited to the online order (and the collection of the groceries at the pick-up point) instead of spending hours in the physical supermarket;<br>- Grocers arrange the goods per storage temperature, which saves time when storing the groceries at home;<br>- When a product is not available, the retailer proposes an alternative. |

Notes: (1) Definitions based on Beauchamp and Ponder [42] and Jiang et al. [43]; (2) Some of the features in the third column can be linked with more than one of the dimensions. We have placed them in the dimension with the strongest fit. (3) Presence and intensity of the features may differ across e-grocers.

As mentioned at the beginning of this Section, for most households the most important convenience aspect of e-grocery shopping is that is saves them a lot of time. In reality, these time savings are not always that clear-cut, but what is important is that consumers *think* that they are saving time [34,44]. For click-and-collect services, Pernot [44], in her study for France, finds that 94% of the users consider that the format saves them time. However, she also observes that users shop for groceries in as many physical retail formats as non-users, implying that users diversify their shopping practices in a greater number of stores. Pernot concludes that the perceived time savings seem to be the result of fragmentation. At the highest level, click-and-collect divides grocery shopping into ordering and collecting, and thus breaks up the act of buying in both time and space [44–46]. Indeed, consumers are no longer subject to opening hours, as ordering can be made at any time and from anywhere. Smartphones—and the possibility to order groceries via the retailer's app—further enhance this ubiquitous "anywhere, anytime" nature of e-grocery shopping [47]. In addition, the ordering process itself can be further subdivided into several stages; for example, one can start ordering during the lunch break at work, continue while returning home, and confirm the order at home in the evening. Finally, the collection of the groceries can often be fitted into existing trips, particularly the home-work commute [48,49].

Overall, the use of click-and-collect services triggers several types of reorganization and, in this way, reduces the spatio-temporal constraints related to grocery shopping [44,50]. The process is turned into a succession of short stages that can be performed at different times and different places, and the time-consuming weekly trip to the supermarket can be fragmented into several trips to several store formats. This twofold fragmentation makes buying groceries a much more flexible activity that can be better integrated in the complex and desynchronized schedules of households. By avoiding interruptions—such as finding (and returning to) a parking space, browsing the shelves to find the right brand, going back to find a forgotten product, or waiting at the checkouts—consumers significantly reduce the amount of time spent doing nothing that is associated with in-store shopping. Click-and-collect shopping allows households to use their time more continuously: each of the fragmented activities can be executed without interruption, so that the feeling of wasting time is reduced. Households thus gain more control over the timing of their grocery shopping, which makes it less of a chore [44]. Along the same line, Gahinet and Cliquet [51] highlight that time convenience comprises a double dimension: "a quantitative dimension (chronos), which refers to clock time and allows 'time saving', and a qualitative dimension (kaïros), which refers to being timely and allows 'time management'." The latter gives consumers the feeling of better managing their time and even controlling it. The authors find that French consumers value both dimensions when visiting (offline) convenience stores. However, in building customer loyalty, *kaïros*, or the feeling of control over their time, proved to be more important than the *chronos* or time saving aspect.

*2.3. Willingness to Pay for E-Grocery Services*

As pointed out in the Introduction, examining the service costs charged by retailers, and, most importantly, consumers' willingness to pay such fees is vital. However, research on the subject is scant [9]. In Table 2, we present an overview of studies that do investigate consumers' WTP for e-grocery services and in the remainder of this section we discuss the key insights.

**Table 2.** Overview of studies on WTP for e-grocery services.

| | Country | Year | Data | Delivery Type | Main Findings |
|---|---|---|---|---|---|
| Kotzab and Teller [4] | Austria | 2002 | Face-to-face interviews with customers of an offline supermarket in the city center of Vienna (*n* = 308). Standardized questionnaire. | Attended home delivery | We found that 25.3% was not prepared to pay anything.<br>In a multiple regression analysis among respondents with a non-zero WTP only 'perceived transportation costs' proved significant. Note that the majority estimated these costs to be zero.<br>No impact of 'shopping frequency', 'basket size' or 'perceived distance'. |
| Teller et al. [11] | Austria | 2005 | Web-based survey among two specific consumer groups: the 'time-starved' (*n* = 384) and the 'new technologists' (*n* = 144). Time-starved: "Dual income household with kids, time pressed" New technologists: "Young and technologically interested, have no time for shopping" | Attended home delivery | The time-starved were prepared to pay a significantly higher fee (modus = EUR 5) than the new technologists (modus = EUR 2).<br>For both groups, significant but weak correlations (of maximum 0.42) were found with 'distance from home to store', 'size of shopping basket', 'average shopping frequency', and 'degree of procurement responsibility' (i.e., the share of the household's grocery shopping done by the respondent).<br>Interestingly, no relationship was found between household income and WTP. |
| Goethals et al. [10] | France | 2009 | Structured survey, face-to-face (*n* = 68) and online (*n* = 177). | Unattended home delivery | The majority had not yet bought groceries online (83%); 32% has WTP of zero.<br>The relationship between distance to the store and consumers' WTP was significantly negative.<br>No impact of 'shopping time' (how long it takes a consumer to grocery shop) and 'in-store shopping pleasure'.<br>Interaction effect between 'in-store shopping pleasure' and 'shopping time' is significant and positive. |

**Table 2.** *Cont.*

| | Country | Year | Data | Delivery Type | Main Findings |
|---|---|---|---|---|---|
| Gil et al. [8] | Europe | 2008–2009 | Transaction data of 29,373 customers (913,842 transactions). | Attended home delivery (consumers pay a time-specific delivery fee, ranging from EUR 4.95 to 11.95). | Delivery fee is positively correlated with basket size and negatively with the number of times the service is used per year. |
| Seitz et al. [12] | Germany | 2012–2013 | Structured survey ($n = 412$), mainly among non-users; 89.8% of the respondents were not even familiar with the concept of e-grocery buying. | Attended home delivery; pick-up; drive-through. | 84.5% was prepared to pay for home delivery, 36.2% for pick-up, and 36.7% for drive-through. Being willing to pay is positively related to 'basket size', 'income', and 'need for convenience'. |
| Milioti et al. [13] | Greece and UK | 2016 | Online questionnaire (Greece: $n = 170$; UK: $n = 367$). Sample consists of consumers that have shopped for groceries online at least once. | Attended home delivery; pick-up from store; pick-up from a locker. (Pick-up from store = reference category) | Home delivery holds a strong position among the distribution modes examined, especially concerning the weekly order, while pick-up from locker can be developed to a competitive alternative for urgent orders in both markets. Greece: in a weekly (urgent) order setting, consumers are willing to pay an additional fee of EUR 3.64 (EUR 3.26) for home delivery and an additional fee of EUR 1.69 (EUR 1.79) for the pick-up from locker option compared to pick up from the store. UK: in a weekly (urgent) order setting, consumers are willing to pay an additional fee of EUR 6.46 (EUR 4.42) for home delivery and an additional fee of EUR 1.65 (EUR 1.97) for the pick-up from locker option. 'Price consciousness' has a negative impact and 'satisfaction with delivery fulfilment' positively influences WTP measures. Time pressure is not significant. |

**Table 2.** *Cont.*

| | Country | Year | Data | Delivery Type | Main Findings |
|---|---|---|---|---|---|
| Brand et al. [52] | UK | 2017 | Online survey among 2032 grocery shoppers. | Attended home delivery | The majority disagreed with the statement 'I would pay more for the convenience of home delivery of groceries' (score of 2.52 on a 1–5 scale). |
| Klepek and Bauerova [53] | Czech Republic | unknown | Web-based survey (*n* = 670). Open question: "Why do you not buy groceries online?" Data collection by IPSOS (online panel). | Not specified | Content analysis identified 'unwillingness to pay for delivery' as one of the thematic units. Theme was ranked 8th out of 14 and frequency analysis showed that 23 out of 945 responses (2.4%) related to consumers' unwillingness to pay. |

A first observation is that the WTP would appear to be rather low. In the majority of the studies, a substantial share of the respondents are simply not prepared to pay a fee. This might be explained by the fact that most surveys relate to a period when online grocery shopping was still in its infancy. Indeed, with the exception of Brand et al. [52], Milioti et al. [13] and Klepek and Bauerova [53], the data stem from 2002 to 2013. As a result, in most studies the number of users in the sample is rather low.

Retailers are not blind to this low WTP. In exploratory qualitative research for Germany, one of the interviewed retailers stated: "German customers are used to paying small amounts of money for groceries and are not willing to pay for any extra services" [54]. This first observation is in line with the findings of research focusing on consumers' preferences concerning last-mile delivery options [55–57]. Several studies conclude that the delivery fee is the most important attribute: higher delivery fees significantly reduce consumers' preference for a particular delivery option.

Second, Table 2 also shows that the majority of the existing studies examine home delivery services. To the best of our knowledge, only Seitz et al. [12] and Milioti et al. [13] investigate the WTP for click-and-collect services. However, the former only asked consumers *whether* they were prepared to pay, and not *how much*. They find that 84.5% was prepared to pay for home delivery, 36.2% for pick-up, and 36.7% for drive-through. Seitz et al. [12] also find that being willing to pay is positively related to basket size and income. In addition, the higher the respondents' need for convenience, the higher the probability that they are willing to pay, which is not in line with the results of Teller et al. [11]; see Table 2. This discrepancy can be explained by the fact that Seitz et al. define 'need for convenience' more broadly: it is measured not only in terms of 'convenience of ordering', and 'independence from opening hours', but also in terms of 'no travel expenses/time loss', 'no queuing', and 'saving time'. Clearly, the latter three dimensions also relate to time pressure.

Milioti et al. [13], in their study for Greece and the UK, use a stated preference ranking experiment to investigate consumer acceptance of three delivery methods as well as their WTP for each of these. Overall, the negative sign of the coefficient related to the attribute 'cost' indicates that higher delivery prices significantly decrease the likelihood of an alternative being ranked first. As for the level of the WTP, both Greek and UK consumers proved to be willing to pay higher fees for the home delivery and the pick-up from a locker option than for pick-up from the store (see Table 2). This is true for both weekly and urgent orders (i.e., same-day delivery). The additional WTP for pick-up from a locker is largest for urgent orders and in the UK.

Milioti et al. [13] also divide their sample in sub-groups based on 'price consciousness', 'perceived time pressure', and 'satisfaction with home delivery fulfilment' (the results are similar for Greece and the UK). They find that the additional WTP for an urgent order is lowest for price conscious respondents—and this for both home delivery and pick-up from a locker. In the weekly order scenario, the WTP measures do not vary significantly between the two groups. Interestingly, the same is true for 'time pressure'. In the urgent order setting, consumers with perceived time pressure are willing to pay slightly more for the home delivery and pick-up from locker options, but the difference is not significant. Conversely, satisfaction with home delivery fulfilment does have an impact. It increases WTP for home delivery, especially in the weekly order scenario, and decreases it for the pick-up from locker option in the urgent order setting.

In another study for Greece, Milioti et al. [15] again investigate consumers' WTP for click-and-collect services. However, this study does not specifically examine e-grocery services but rather e-commerce in general. In addition, Milioti et al. only asked consumers whether they were prepared to pay, and not how much. Although the majority of respondents stated that they would use a click-and-collect service, only 28.4% indicated that they would be willing to pay for it. The binary logistic regression model shows that 'environmental perception', 'low income', and 'perceived time pressure' significantly impact consumers' intention to pay for a click-and-collect service. Where the latter variable is concerned, the model indicates that those who feel they do not have enough time to do their grocery shopping are approximately three times more likely to pay for the service.

A final observation is that, as mentioned in the Introduction, the literature on online grocery shopping suggests that retailers should price discriminate by asking higher fees for popular and/or rush-hour time slots. Alternatively, Brand et al. [52], in their recent study for the UK, suggest the use of explicit versioning. One of the five grocery shopper segments that they identify are the 'intensive urbanites'. Concerning this segment Brand et al. [52] (p. 13) remark: "As they include the highly connected, affluent, busy, and convenience-loving shoppers who are prepared to pay more for the convenience and ease of use, there is the potential for a separate, super-convenient service to be paid at a premium (e.g., an 'Amazon Prime' for groceries)". However, hardly any authors test propositions like these from the customer point-of-view. In fact, only few studies can link time pressure with WTP, either directly or indirectly. The main goal of the present paper is therefore to explicitly analyze whether time-pressed consumers are willing to pay higher fees. In the following section we proffer theoretical underpinnings for this proposed relationship, and further develop our conceptual framework.

## 3. Conceptual Framework and Hypotheses

Time in general and feelings of time pressure in particular have been acknowledged to be crucial factors in the complex realm of consumer decision making [58–60]. However, in household shopping decision making and behavior, time is the factor that has been studied the least [61]. In the grocery retailing context, a few studies have examined the impact of perceived time pressure on consumers' behavior in (offline) supermarkets [62–64] and on their adoption of online grocery services [65–67]. Where the latter is concerned, only three of the studies discussed earlier (see Table 2) can (indirectly) link time pressure with WTP.

Teller et al. [11] find that time-starved consumers have a higher WTP than 'new technologists'; however, their study is limited to these two consumer groups. Note that alumni from the Vienna University of Economics and Business Administration represented the 'time starved', while current students represented the 'new technologists'. This would seem imperfect proxies for how Teller et al. theoretically define the two groups (see Table 2). Teller et al. also do not control for differences in income between the two groups.

Seitz et al. [12] find that consumers' 'need for convenience'—which includes 'saving time'—is positively linked with the probability to be willing to pay. However, their survey only queried respondents about whether they were prepared to pay, and not about the level of their WTP. In the same vein, Milioti et al. [15], for e-commerce in general, conclude

that time-pressed consumers are more likely to be willing to pay to collect their order from a click-and-collect point. As an aside, it can be noted that Melis et al. [68] (not discussed in Table 2) find, for the case of the UK, that consumers' decision to start ordering groceries online at a particular supermarket chain leads to an expansion of the share of wallet assigned to that chain. Interestingly, this expansion is significantly stronger for the time-constrained. (Consumers' time constraints were approximated by the number of shopping trips they made before they started shopping online, with a lower purchase frequency implying higher time constraints.) This suggests that time-starved consumers become more 'locked in' to the online channel and might thus have a higher WTP.

Despite this lack of explicit evidence, the hypothesis that consumers who are pressed for time have a higher WTP makes sense in theory. Building on the classic economic choice model, Becker [69] already acknowledged that time, just like income and prices, constrains choice, leading him to present his time allocation theory. The key assumptions are that households are not only consumers but also producers, and that time—a scarce resource—is used in both activities. Hence, households rely on the cost-minimization rules of the traditional theory of the firm as they produce basic commodities by combining market goods and time. The quantities of such commodities that are produced are then determined by maximizing a utility function under time and budget constraints.

Researchers relying on these economic household production models view time as an opportunity cost of forgone income or participation in other activities. In other words, time and money are, at least partly, interchangeable [70–72]. For some commodities (such as cleaning or grocery shopping), households can decide whether to invest time (and perform the activity themselves) or invest money in order to acquire a good or service that can (partly) execute the task for them [73]. Generally, it can thus be assumed that those with a high opportunity cost of time (measured in terms of income or hours worked) will perceive services with a time-saving and/or time convenience component as particularly attractive.

Transposed to the setting of the present paper, one would expect that consumers with higher levels of perceived time pressure are more sensitive to the time costs of grocery shopping, so that they value the convenience and time-saving aspects of click-and-collect services more highly. They should thus be willing to pay more for such services. In what follows, we discuss the existing empirical evidence on this reasoning.

In the marketing literature, several studies have investigated the factors that influence consumers' use of convenient goods and services. Next to demographic variables such as 'household income' and 'stage in family life cycle', lifestyle variables that have been found to correlate with convenience consumption include 'role overload' and 'time pressure' [58,74–76]. Furthermore, ample research—in a variety of contexts and countries—revealed a link between consumers' perceptions of (the different dimensions of) service convenience and their overall evaluation of a service in terms of, for example, increased satisfaction and enhanced loyalty [77–80].

The conceptual literature on service convenience also suggests that consumer characteristics (including perceived time pressure) moderate the relationship between convenience and satisfaction [40]. Still, only few studies empirically explore such propositions. Benoit et al. [77], in a study for Western Europe, find that the impact of service convenience on customer satisfaction with a grocery store is stronger when time pressure—measured as respondents' perceived constraints on time available for shopping—is high. In the same line, Kleijnen et al. [81] (p. 37) investigate the moderating effect of time consciousness—defined as "the extent to which consumers are aware of the passing of time and how they spend it"—on the value creation process in the context of mobile services. (Note that the scale for time consciousness includes the item "I usually feel pressed for time".) The authors find that for consumers with higher time consciousness the positive relationship between time convenience and value is stronger than for those with lower levels.

Besides the antecedents of customer satisfaction, researchers have also investigated the consequences of a positive evaluation of a service and its positive impact on consumers' WTP [82]. Goebel et al. [83], for example, first address the antecedents of perceived

attractiveness for a convenience-enhancing service—the time-based delivery of parcels—and subsequently explore how consumers' perceptions on the attractiveness of the service affect their WTP for it. For the case of Germany, the authors find that 'availability at home to receive a parcel' negatively influences perceived attractiveness, whereas 'amount of time required to pick up a parcel at the office of the logistics provider' has no impact. In addition, they show that consumers' perceptions of attractiveness are positively related to 'the number of hours a consumer works per week'—which can be considered to be a proxy for time pressure. In other words, consumers do seem to take the opportunity cost of time into account, and those with a higher workload find convenience-enhancing services more attractive. Finally, when examining the outcome variables, Goebel et al. [83] find that consumers with higher levels of perceived attractiveness have a higher WTP.

Several other studies in the online shopping literature have examined similar relationships. In an early study for the US, Srinivasan et al. [84] demonstrate that e-loyalty has a positive impact on consumers' willingness to pay more. Surprisingly, of the eight identified antecedents of e-loyalty, only 'convenience' was not significant, which is not in line with the hitherto cited studies. Note that Srinivasan et al. [84] use a narrow definition of convenience, namely, "the extent to which a customer feels that the website is simple, intuitive, and user friendly". Herhausen et al. [85] find that German consumers' perceptions on service quality of an Internet store positively influence their WTP. In a recent study for Bangladesh, Saha et al. [86] show that consumers' WTP is enhanced by their perceptions on delivery efficiency, and that this relationship is mediated by customer satisfaction. Pham and Ahammad [87], however, find no relationship between online customer satisfaction and willingness to pay more.

Other studies explicitly link higher WTP with higher levels of time pressure. Etkin et al. [88] (p. 396), for example, suggest that "feeling time constrained should make consumers value their time more highly. As a result, they should be willing to pay more to save time". The authors first show that perceptions on goal conflict — present when the pursuit of one valued goal hinders the pursuit of another valued goal [89] — influence how time constrained consumers feel. In turn, by inducing how customers value their time, perceived goal conflict then appears to impact consumers' WTP to have things done faster. In one of their experiments, Etkin et al. [88] (p. 404) find that "participants who perceived more goal conflict were willing to pay approximately a 30% shipping premium to receive their order sooner".

Based on the above reasoning, we propose the main hypothesis of our study:

**Hypothesis 1 (H1).** *Time-starved consumers have a higher WTP for online grocery services than consumers who are less pressed for time.*

In our study, we define perceived time pressure as "the degree to which an individual considers him- or herself busy, so that shopping causes high opportunity costs and a feeling emerges of not having enough time to make good purchases" [90] (p. 150). As explained by Settle and Alreck, this "degree to which one perceives oneself as lacking time relative to the daily tasks of living—can be seen to arise from two, fairly distinct sources: One is situational, the other, personal" [91] (p. 86). That is, people can have the impression that they have too many things to do and too little time to do them. Such situational sources of time pressure result from "circumstances imposing demands and impinging on one's time *involuntarily*" [91] (p. 86). Those who are experiencing personal time pressure, on the other hand, "put themselves there by choice or inclination" [91] (p. 86). In other words, they routinely take on more and more tasks and accept obligations without considering whether they will have the time to satisfy all these commitments.

H1 and the accompanying reasoning above applies to both sources of time pressure. Those experiencing situational time pressure will mainly appreciate click-and-collect grocery services because of the time-saving aspect. For those whose personal inclinations result in time pressure, time convenience will be more appealing [91]. That is, they will

mainly value the fragmentation aspect of click-and-collect. They use the service because it allows them to engage in polychronic activities, which enables them to do more tasks quickly. For both types of consumers, it can thus be postulated that their feelings of time pressure will increase their perceptions on the attractiveness of the service, which, in turn, will increase their WTP for it.

As for the determinants of time pressure, the impact of (household) income has been well explored [72,92–95]. The literature on heuristic decision making and that on opportunity costs both suggest that the economic value or worth of one's time can influence the perceptions of time pressure. The reasoning stems from the association between value and scarcity, which has long been examined in one causal direction: the scarcity of something enhances its value [94]. However, at a certain point in time this relationship was turned around and it was argued that if people perceive something as valuable, they also assume it to be more scarce [96,97]. In other words, higher economic value of time increases feelings of time pressure [94]. Indeed, greater income increases consumers' opportunity cost or value of their time, which leads to higher perceived time pressure [72].

In addition, since income does not give information about the working situation of all household members, employment status should also be taken into account. Indeed, it is possible that household income is high because one of the partners has a well-paid full-time (or more than full-time) job, but that the other partner works only part-time or even not at all. In such a situation, the latter will probably be in charge of the grocery shopping. We therefore think that it is better to capture time pressure by looking into the employment situation of both partners, in parallel with (household) income. In a study for Belgium, Van Droogenbroeck and Van Hove [33] find that the time-saving motivation is positively related to the *relative* number of full-time employed adults in the household. The logic is that many households where all adults work full-time have little choice but to shop in the evening or weekends. As these consumers value time saving the most, their WTP for an online service might be higher as well. Besides, in general, longer hours of work are associated with a sense of time pressure [64,92,94].

Prior studies also acknowledge the impact of age, education, and presence of (young) children on feelings of time pressure. For age, the literature finds that those feeling most pressed for time are the (relatively) young and that perceived time pressure diminishes as one reaches later stages of life [64,92,94,98]. Where education is concerned, the findings are less clear-cut. While some studies [64,98] show that time-poor consumers tend to be well-educated, DeVoe and Pfeffer [94]—for the case of Australia—find that obtaining a bachelor's degree or higher is negatively related with the experience of time pressure. In their study for Korea, Cha and Suh [92] find that the well-educated are more likely to feel rushed. However, at the same time, Cha and Suh's longitudinal analysis revealed that, as they become older, those who are more educated experienced a huge drop in perceived time pressure compared to those with lower levels of education. Finally, the household-level characteristic 'presence of children' also tends to influence time pressure. That is, those living in household with young children demonstrate higher levels of perceived time pressure [64,92,94,99].

To sum up, where the antecedents of perceived time pressure (PTP) are concerned, we propose the following hypotheses:

**Hypothesis 2a (H2a).** *The higher the household income, the higher the level of PTP.*

**Hypothesis 2b (H2b).** *The higher the relative number of full-time employed adults in the household, the higher the level of PTP.*

**Hypothesis 2c (H2c).** *The younger the consumer, the higher the level of PTP.*

**Hypothesis 2d (H2d).** *The higher the consumer's education, the higher the level of PTP.*

**Hypothesis 2e (H2e).** *The level of PTP is higher for those living in households with young children than for those living in households without young children.*

Looking back at the studies that have already explored the WTP for e-grocery services (see Table 2), an observation is that consumers' (dis)like of grocery shopping appears to have no impact on their WTP for the online channel [10,11]. This might be explained by the fact that in the samples under investigation, respectively, 78.9% and 83% were consumers without experience in e-grocery buying. In the more recent qualitative study of Van Droogenbroeck and Van Hove [34] several users stated that they did not like the crowdedness in physical supermarkets, the waiting time at the checkout, and the overabundance of brands and products, and that this—amongst other reasons—had incited them to start ordering groceries online. Picot-Coupey et al. [32], in their study for France, confirm this impact of the dislike of grocery shopping. In short, the relationship between perceived in-store shopping enjoyment (PSE)—or rather the lack thereof—and WTP would seem to exist after all. We thus propose the following hypothesis:

**Hypothesis 3 (H3).** *Consumers with a higher level of PSE have a lower WTP than consumers with a lower level of PSE.*

Another observation is that the most frequently investigated variables relate to consumers' logistics costs. However, here a problem is that off- and online shopping frequency and basket size are, in our view, not sufficiently comparable. It is perfectly plausible that a consumer who, prior to adoption of an e-grocery service, goes to the supermarket three times a week changes to, say, two-weekly online orders. Moreover, the relationship between online basket size and WTP may well be bi-directional. Users with a relatively low WTP might aggregate their purchases more, so as to reduce the number of times they have to pay the service cost. We therefore only include multichannel behavior in our model—and obviously only for users. Users who still visit physical supermarkets on a regular basis are less tied to the online channel and incur lower switching costs. Their WTP might therefore be lower than for consumers who visit a physical store only when they have forgotten something. Note also that including distance to the store makes little sense in a pick-up context. For the users in our sample we posit that:

**Hypothesis 4 (H4).** *Multichannel consumers will have a lower WTP than consumers who visit physical stores only when they have forgotten something.*

Finally, we also investigate the direct impact of (household) income on consumers' WTP. Several studies in the broad literature on shopping for grocery products confirm its positive influence [100,101]. Hence, we hypothesize that:

**Hypothesis 5 (H5).** *The higher the household income, the higher the WTP.*

Figure 1 summarizes the above observations and hypotheses. In essence, we hypothesize that both personal- and household-level variables determine the feeling of time-poorness, which in turn has a direct effect on WTP. In addition, we test the direct effects of 'perceived shopping enjoyment' and 'household income'.

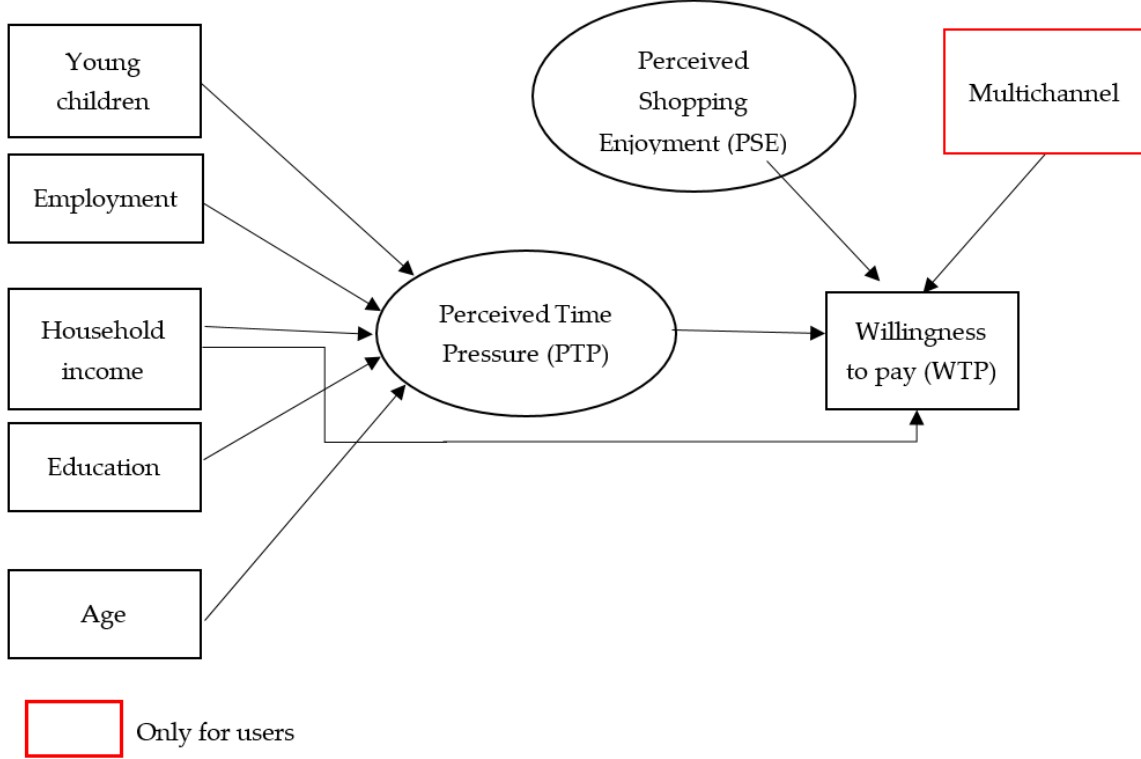

**Figure 1.** Conceptual model.

## 4. Methodology

This section first discusses the data collection. The second part explains the operationalization of the variables.

### 4.1. Sampling Strategy and Data Collection

To collect our primary data, we conducted a survey among the customers of two Belgian supermarkets, Colruyt and Delhaize. In doing so, we limited ourselves to Brussels and Flanders (the Dutch-speaking part of Belgium), and we distributed the questionnaires at specific stores and pick-up points.

The locations were selected based on three geographical criteria: region, level of urbanization, and presence of supermarkets and pick-up points. Region is based on the Nomenclature of Territorial Units for Statistics (NUTS), level 2; the level of urbanization is based on the urban hierarchy of Loopmans et al. [102]. We first made a list of Delhaize and Colruyt supermarkets that also had a click-and-collect pick-up point (in Belgium these pick-up points are usually physically attached to the supermarket, but not all branches have one). The rationale was to make sure that the offline and online channel were (roughly) equally attractive in terms of location, so that we could eliminate this variable from our model. From this list we then randomly selected one urban and one suburban location in each of the five Flemish provinces. In addition, one (by definition urban) location was chosen in Brussels.

The questionnaires—which were available in Dutch or French—were distributed at the 11 locations between 17 November and 13 December 2014. In order to not disturb the customers while shopping, they were approached when they were leaving the store or the pick-up point. In order to ensure the presence of both working and non-working consumers in our sample, the data collection was performed on both weekdays and Saturdays, and at different moments during the day (on average 8 h per location). Respondents could either fill in the questionnaire on paper (and mail it back in a stamped and addressed envelope), or they could complete the survey online (a link to the online survey was mentioned on the paper copy). Note that the instructions specifically asked for the household member

primarily responsible for the grocery shopping to complete (the bulk of) the survey. As an incentive we organized a raffle, in which ten respondents could win a EUR 25 voucher for a supermarket of their choice. In total, we distributed 2382 questionnaires, with an average of 213 copies per location. The total number of questionnaires completed was 638 (or 26.8%). After data cleaning, the total number of valid responses was 572: 105 users and 467 non-users.

### 4.2. Operationalization of the Variables

When one wants to measure consumers' WTP for a good or service, a first question is whether one should try to determine their hypothetical or actual WTP. In the latter case, one can use the Becker–DeGroot–Marschak mechanism. In this set-up, participants are obliged to purchase a product if the price drawn from a lottery is less than or equal to their stated WTP [103]. The idea behind such a lottery is that consumers will reveal their 'true' WTP, as they have no incentive to state a higher or lower price. This approach is often used for new products but does not seem suited for our purposes. Especially for the users in our sample, it might lead to underestimates. As users know the real service cost, they have no incentive to state a higher price because then they risk being obliged to use the service at a higher price than the market price. Some of the non-users might also be aware of the level of the service cost.

We therefore chose to try to determine consumers' hypothetical WTP. There are two approaches to do this. In the direct approach, consumers are asked to state their WTP for a specific product or service through, for example, an open-ended question (OE). Alternatively, in the indirect approach, WTP is calculated based on consumers' choices among several product alternatives, via, for example, choice-based conjoint (CBC) analysis.

As in all but one of the existing e-grocery studies, we opted for the direct OE method. We know from our literature review that the willingness to use (WTU) and the WTP for online grocery services is rather low, and that several consumers decide to 'opt out'. If we were to use CBC, we would thus have to include a no-choice option in order to increase the similarity with the real-life situation. However, adding such an option has its drawbacks. For example, if there are many no-choice outcomes, the number of observations that can be used to detect utility differences between attribute levels is reduced [104]. Our choice for OE is also justified by Miller et al.'s [103] comparison of WTPs derived by means of the four commonly used approaches with real purchase data. Miller et al. find that while actual WTP methods result in more reliable measures than both OE and CBC, OE can outperform CBC in estimating WTP for inexpensive, frequently purchased, nondurable product categories.

As for the phrasing of our OE questions, for the *non-users* we opted for a two-stage design. Non-users were first asked the following question: "If you could use an online grocery service at a price that you consider acceptable, would you do so?" [105]. Participants who responded positively were subsequently requested to indicate an acceptable and a maximum service cost. Note that at the time of the data collection, for Collect & Go the service cost was a fixed amount of EUR 5.50 per order (in May 2020 this was increased to EUR 5.95). For Delhaize.be the fee was EUR 4.50 (the same as at the time of writing). (Delhaize did briefly experiment with a variable service cost, depending on the day the order was collected. The fee was EUR 3 on Tuesday, Wednesday, and Thursday, and EUR 4 on Monday, Friday, and Saturday.)

The *users* in our sample were queried about their behavior in a scenario where the service cost would increase. In particular, we asked them: "At which level of the service cost would you completely stop using online grocery services and only visit physical supermarkets?" The answer to this question is taken to represent their maximum WTP. In addition, we asked: "At which level of the service cost would you adjust your usage of online grocery services (for example order less frequently)?" We call this the critical price point.

With the above information we constructed three alternative WTP measures, as we do not know for sure just how consumers gauge the level of the service cost. The first is the maximum WTP in absolute terms (for both users and non-users); that is, the maximum service cost (in euro, per order) they are willing to pay (WTPABS). Note that for the non-users without a willingness to use we simply set the maximum WTP at zero. We also calculated a relative maximum service cost (WTPREL), as well as a maximum total cost per year (WTPTOT)—for the users, that is. The former measure expresses the maximum WTP as a percentage of the user's average amount per order. For the latter we multiplied the critical price point (in euro, per order) with a factor related to the frequency of use. This factor was set equal to 60, 52, 26, 12, or 8 for, respectively, users who indicated that they used the service 'more than once a week', 'once a week', 'once every 14 days', 'once a month, and 'less than once a month'. Note that to compute the maximum yearly cost, it would not be realistic to multiply the (current) order frequency with WTPABS because this is the price point at which the respondent would simply stop ordering online. This said, WTPTOT, in the way we compute it, is also but a proxy. Indeed, given that the critical price point is the level of the service cost at which the respondent would start adjusting her behavior, the corresponding order frequency will be lower than the current (the level that we use in our calculations). However, since we do not know the precise reaction path of the respondents between their critical price point and their WTPABS, we cannot determine precisely when the multiplication of service fee and order frequency will reach its maximum. WTPTOT may thus well overestimate the maximum yearly cost for some respondents and underestimate it for others.

Turning to the independent variables, 'presence of young children' was coded 1 when there were children younger than 11 years old in the household, and 0 otherwise [64]. The relative number of full-time employed partners was calculated as the number of full-time working partners (0, 1, or 2) divided by the total number (1 or 2) in the household. This variable is not entirely the same as in Van Droogenbroeck and Van Hove [33], who work with the total number of adults in the household. The reason is that while we know the number of adults in the household, we only have data about the employment status of the respondent and his or her partner, if any. Income was queried in categories in the survey. Afterwards we used the midpoints of these categories as proxies for individual income, and—if applicable—calculated household income by adding up the income of the respondent and the partner. The income of adults other than the respondent and his/her partner was not questioned. For these cases we thus underestimate household income (30 cases). This is only a problem to the extent that the other adults—an adult child or an elderly parent, say—contribute to the household budget. Note that we also tested our model using the income of the respondent rather than household income. Unless mentioned otherwise, these analyses yielded similar results. The respondents' age was measured as a continuous variable; education is a categorical variable.

To continue with the variables that are hypothesized to determine consumers' WTP, the multi-item scale for the construct 'perceived time pressure' (PTP) was adapted from Van Kenhove and De Wulf [64] and Verhoef and Langerak [67]. The scale for 'perceived in-store shopping enjoyment' (PSE) was drawn from Spangenberg et al. [106]. Both scales can be found in online Appendix A. The items were measured using a seven-point Likert scale, ranging from 'strongly disagree' (−3) to 'strongly agree' (+3). Finally, the variable 'multichannel' was coded 1 if the respondents indicated they still visited physical supermarkets on a regular basis, and 0 if they no longer shopped offline for groceries or only when they had forgotten something.

## 5. Descriptive Results

This section discusses the composition of our sample in terms of individual- and household-level characteristics, the respondents' grocery shopping behavior, and their WTP for online grocery services.

*5.1. Composition of the Sample*

Our sample comprises 105 users and 467 non-users of Belgian online grocery services, of which 70.1% are female. The dominance of women is probably the result of our call to have the questionnaire filled out by the household member most often responsible for grocery shopping, combined with the fact that grocery shopping is still mainly performed by females [107]. The respondents' age ranges between 18 and 86, with an average of 49 (SD = 15). The users are significantly younger than the non-users (41.6 vs. 50.8 years; $t(214) = 7.411, p \leq 0.001$). The majority (61.9%) of the users are in the 25–34 or 35–44 age category, while the greater part of the non-users (69.2%) are 45 or older. The majority of the respondents hold either a professional bachelor's (35.7%) or a university degree (24.1%); 44.9% work full-time, 21.7% have a part-time job, and 23.1% are retired. The relative number of full-time employed partners is zero in 35.1% of the households. In 25.7% half of the partners work full-time, in 37.1% all partners have a full-time job (2.1% is missing). The share of households where all partners work full-time is significantly higher in the users sample (55.8%) than in the non-users sample (33.8%); $\chi^2$ (2; $n$ = 560; 33.950, $p \leq 0.001$). Where net income is concerned, the most frequently cited category is EUR 1500–1999 (26.2%). The household income ranges between EUR 250 and 12,500, with an average of EUR 3486 (SD = EUR 1784).

Looking into the household-level characteristics, household size ranges between one and six, with an average of three (2.6, SD = 1.1). The users live in larger households than the non-users (3.2 vs. 2.4; $t(568) = -6.410, p \leq 0.001$). This is because the share of households without children is significantly higher among the latter (75.9% vs. 41.0%); $\chi^2$ (3; $n$ = 562; 58.160, $p \leq 0.001$). Among the households with children, the age of the youngest child is significantly lower for the users (six years) than for the non-users (nine years); $t(170) = 2.370$, $p = 0.019$. Finally, about half of the respondents reside in an urban area (55.1%); 83.2% is Dutch-speaking. Almost all participants have access to at least one car (90.0%).

*5.2. Grocery Shopping Behavior*

Zooming in on the users in our sample (18.4%), a first observation is that they are clearly overrepresented. At the time of our data collection (in 2014), only 9% of the Belgian consumers who bought goods or services via the Internet had already used an online grocery service [6]. The majority of our users (50.5%) had between one and five years of experience with ordering groceries online, whereas 15.3% only started less than six months ago.

The most popular order frequency is 'once every 14 days' (28.6%), followed by 'once a week' (25.7%), and 'less than once a month' (22.9%). The largest proportion of respondents (37.1%) spend between EUR 100 and 149 per order; 26.7% order for more than EUR 200. In line with the literature [108], most users (65.7%) are multichannel shoppers who combine online orders with regular visits to physical supermarkets. Only 2.9% claim to have stopped buying groceries offline, while 30.5% only go to a physical store when they have forgotten something.

Turning to the picking-up of the groceries, four out of five (80.8%) have either a fixed day and/or a fixed time slot, almost half of the users (48.5%) have a fixed moment (fixed day *and* time slot). 52.6% have a fixed day but not necessarily a fixed time slot; 76.8% have a fixed time slot but not necessarily a fixed day. Saturday between 08:00 and 13:00, and Friday between 17:00 and 21:00 are the most popular time slots; see Table A1 in Appendix. Overall, one out of four (24.2%) collect their groceries on Friday afternoon/evening or on Saturday.

Finally, an important observation is that the most preferred service is Collect & Go, the online service of supermarket chain Colruyt. Almost 90% of the respondents (89.5%) indicate that Collect & Go is (one of) the service(s) they use most often (in terms of number of orders). The behavior of our sample thus mainly relates to Collect & Go shoppers. In other words, our findings are not necessarily valid for the total population of users of online grocery services.

The popularity of Collect & Go is not that surprising. Colruyt is the Belgian pioneer in online grocery shopping. The other large supermarket chains (such as Delhaize and Carrefour) only recently started to further expand their online channels. In August 2016, Delhaize integrated its home delivery service 'Caddyhome' with its click-and-collect service 'Delhaize Direct'. Via 'Delhaize.be' consumers can now order online and choose between home delivery, picking up in-store, or picking up at a drive-in point [109]. Meanwhile, Carrefour is in the process of expanding its drive-through concept and has started a partnership with Bpost, the Belgian postal service, in order to further develop its home delivery model [110]. Overall, click-and-collect is even today still the dominant model in Belgium. Together with France, Belgium is thus something of an outlier in the Western-European context. In countries such as the UK and the Netherlands home delivery is dominant.

*5.3. Willingness to Pay for Click-and-Collect Services*

We found that 18.0% of the non-users responded positively to the question, "If you could use an online grocery service at a price that you consider acceptable, would you do so?" (and were then asked to indicate both an acceptable and a maximum service cost). This suggests that for 82.0% of the current non-users, the value proposition is not attractive enough at any price and/or that inhibiting factors other than price are keeping them from starting to order online.

When exploring the fee levels provided by non-users, it turned out that six respondents who indicated they would be willing to use an online grocery service at an acceptable price did not, on closer scrutiny, have a WTP. In particular, they indicated zero for both the acceptable fee (which is possible; see below) *and* for the maximum service cost (which does not make sense). They were thus excluded from our further analysis. Six other respondents with an acceptable fee of zero were retained as they did specify a non-zero maximum service cost. Remarkably, a group of respondents who indicated that they would not start ordering online at an acceptable price still indicated a reasonable and/or maximum fee (14 cases). As these respondents might have misinterpreted the questions concerning WTP, we dropped these observations. We also decided to omit respondents with fees above EUR 20—for both users and non-users—as this was the maximum amount found by other studies [10]. The total sample size for further analysis is thus 543 (445 non-users and 98 users).

The maximum WTP per order for the total sample averages EUR 2.69 (SD = 4.68, $n$ = 542; one respondent indicated an acceptable fee but no maximum cost). As a reminder, for the non-users without a willingness to use, we set the maximum WTP at zero.

Figure 2 shows the absolute WTP per order for the non-users with a non-zero WTP ($n$ = 76). The acceptable fee ranges between EUR 0 and 10, with an average of EUR 3.54 (SD = EUR 2.09); the breaks in the gray curve are caused by missing values (5) for the acceptable fee. The maximum fee ranges between EUR 1 and 15, with an average of EUR 5.31 (SD = EUR 2.45); when respondents indicated a range, we took the midpoint (two cases). Interestingly, for both fees the modus is EUR 5 (38.2% and 46.1%, respectively), which is just below the EUR 5.50 fee of Collect & Go and even above the fee charged by Delhaize and other Belgian pick-up services. On the left-hand side of the graph, it can be seen that some respondents indicated a maximum fee above EUR 5.50 (20 cases, or 26.3%); three respondents even stated an acceptable fee above EUR 5.50. These observations suggest that many non-users were not aware of the exact level of the service cost and/or that other inhibiting factors were stopping them—even though they implicitly (in the first-stage question) indicated that this was not the case. Figure 2 also shows that for a substantial share of the subsample under consideration (38.2% or 29 cases) the acceptable fee is the same as the maximum fee. The average difference between the two fees equals EUR 2.0 (EUR 3.3 if we drop the differences of zero).

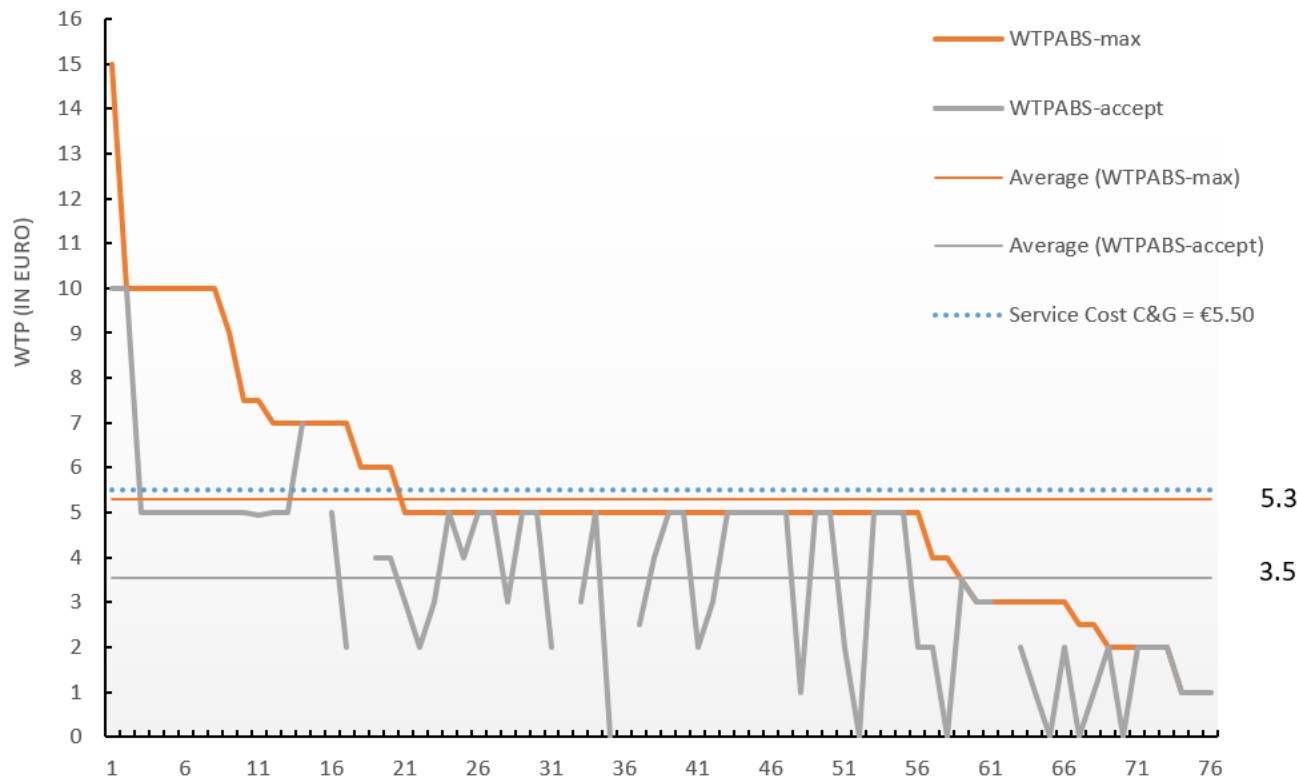

**Figure 2.** WTPABS for the non-users with a willingness to use, in descending order.

Turning to the users of click-and-collect services (*n* = 98; see Figure 3), their maximum WTP—the service cost at which they would stop—ranges between EUR 2.5 and 20, with an average of EUR 10.8 (SD = EUR 4.3). The modus is EUR 10. On average, the users would change their behavior at a fee of EUR 8.1 (SD = EUR 3.3). The range of the critical price point is between EUR 2 and 20, and the modus is again EUR 10. For more than one fifth of the respondents (22 cases or 22.4%), the two price points coincide. These users would thus continue to order groceries online with the same frequency until a certain point, after which they would stop completely. Overall, the difference between the two price points is EUR 2.8 (EUR 3.6 when omitting the differences of zero). Figure 3 also shows that the differences are larger when the maximum WTP is above the average of EUR 10.8. Note that a few respondents indicated fees below EUR 5.50, which, at first sight, is odd as these respondents would seem to have a lower maximum WTP or critical price point than the fee they are paying. However, for users of services other than Collect & Go the fees are lower. Moreover, the fee for Collect & Go used to be EUR 4.50 until 2011 and then changed to EUR 5.50. The Collect & Go users who indicated EUR 5 might not know exactly what the service cost is, they only know that it is "around EUR 5".

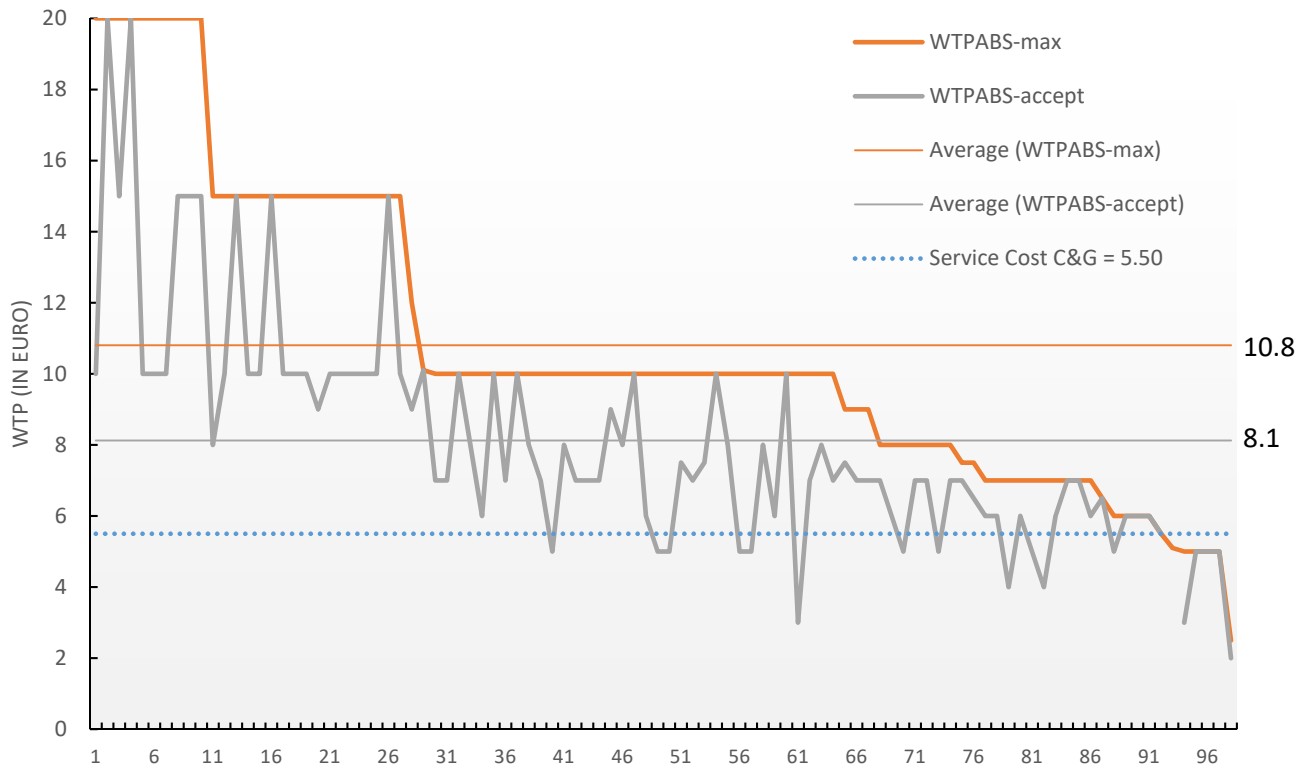

**Figure 3.** WTPABS for the users, in descending order.

### 6. Regression Results

In order to estimate the relationships put forward in our conceptual framework in Figure 1, we applied a system of regression equations. First we explain PTP by means of a multiple linear regression analysis comprising both personal-level (age and education) and household-level characteristics (income, employment, and presence of young children) as independent variables. The predicted values for PTP are then used in the regression for WTP. All analyses are performed with the statistical package STATA, version 14.

We examined boxplots and conducted independent sample t-tests and one-way analysis of variance. Additional preliminary analyses show that our measurement scales have acceptable reliability, convergent validity, and discriminant validity. Except for PSE and household income the incidence of missing data in our sample is low (levels well below 5%). It was therefore safe to use the simple mean imputation method to replace them. For PSE, we applied a variant of the two-way imputation method [111]. In particular, we took the average of the person mean and the relevant (user or non-user) group mean. For household income we used latter. Moreover, our data do not suffer from multicollinearity. All tests are available upon request from the lead author.

Preliminary tests showed that, for the full sample, the explanatory variables individually have an impact on the dependent variables. However, when we repeated the analyses, first, for the sample of respondents who have a positive WTP and, second, for the sample of users, several relationships disappeared. For example, in the users sample there is no relationship between PTP and WTPABS (see Figure 4). The disappearance of several relationships is probably linked with the presence, in the analyses for the full sample, of a relatively high number of respondents with a WTP of zero.

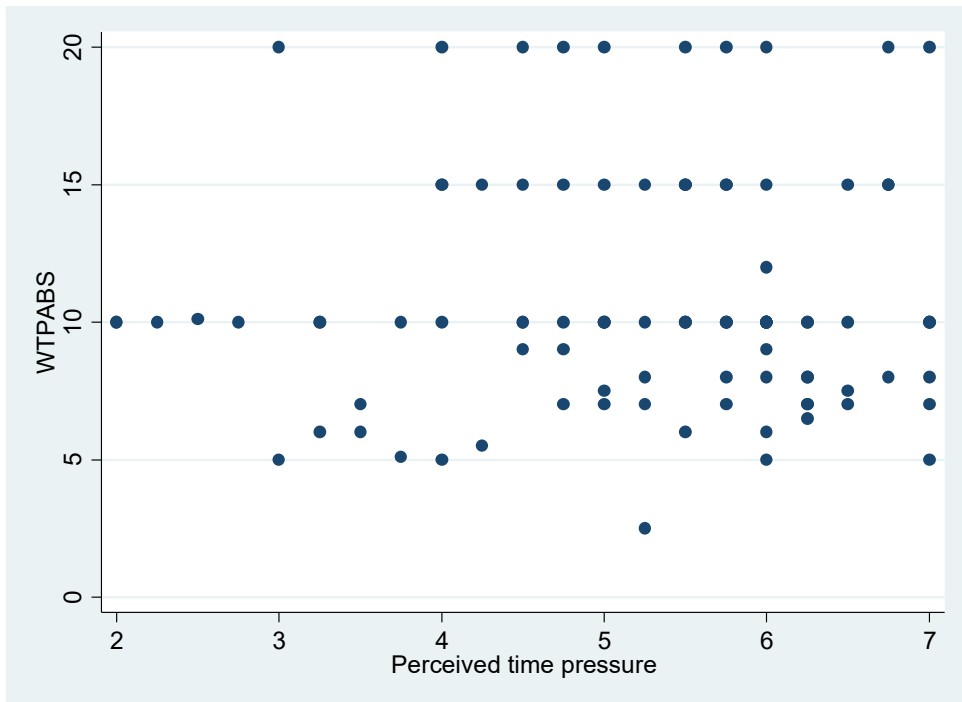

**Figure 4.** PTP vs. WTPABS (users only).

In our actual analysis we therefore follow a three-step approach. In particular, we perform the regression analyses for three samples: (1) all respondents, (2) respondents with a non-zero WTP, and (3) respondents who currently use click-and-collect services. For the latter, we test three alternative models with, respectively, absolute maximum WTP, relative maximum WTP, and maximum yearly cost.

An important methodological remark is that because working with non-random subsamples introduces a risk of selection bias, we use a two-stage Heckman procedure for steps 2 and 3 of our three-step approach. The intuition is that, say, the subsample of respondents with a non-zero WTP may have specific characteristics that influence the fact that they are willing to pay for a click-and-collect grocery service. Therefore, in a first stage (the selection equation) we explain whether respondents are willing to pay and only in a second stage (the outcome equation), how much they are willing to pay. The Heckman procedure assumes that the decision to pay and the decision on the amount that one is willing to pay occur simultaneously, and the procedure involves an estimation in two steps, namely, selection and response. Similar remarks hold for step 3. The Heckman model entails the estimation of the following two equations:

$$y_i = X_i'\beta + \varepsilon_i \tag{1}$$

$$z_i = W_i\gamma + \mu_i \tag{2}$$

where $z_i$ is a binary variable and $y_i$ is only observed when $z_i = 1$; in other words, when respondents have a non-zero WTP (in step 2) or when they are users of an online grocery service (in step 3). Equation (2) is the identification or selection equation and is a probit-type equation, while Equation (1) is the response or outcome equation, which shows the relationship between the predictors $X_i$ (PTP, PSE, household income, and multichannel) and the amount that respondents are willing to pay $(y_i)$.

The Heckman model is estimated using full information maximum likelihood. The choice of variables in the selection equation is based on earlier research by Van Droogenbroeck and Van Hove [33] on the adoption of online grocery services in Belgium and includes: education—the highest level of education of the respondent; age—the age of

the respondent; children—presence of young children, coded 1 when there were children younger than 11 years old in the household, and 0 otherwise; employment—the relative number of full-time employed partners, calculated as the number of full-time working partners (0, 1, or 2) divided by the total number (1 or 2) in the household; and income—net monthly household income.

### 6.1. Full Sample

Our results for the full sample can be seen in Table 3. Where PTP is concerned, a first observation is that all demographic factors have a significant effect, albeit for household income only at the 10% level. Unsurprisingly, households with young children feel more pressed for time than those without. A higher relative number of full-time working partners also increases the feeling of time-poorness, as does education and household income. Conversely, PTP decreases with age.

Second, in line with our hypothesis, PTP (as predicted by the first equation) is a predictor of WTPABS. Consumers who are pressed for time have a higher WTP for click-and-collect services. Third, household income also has a positive impact. (As explained in the Methodology section, we also tested our model using 'income of the respondent'. In this analysis, the relationship between individual income and WTP is not significant.) Conversely, the higher the perceived enjoyment of grocery shopping in physical stores, the lower the WTP for e-grocery services.

**Table 3.** Results for the full sample.

| DV: PTP (*n* = 543) | Unstandardized Coefficients | | Standardized Coefficients | t | *p*-Value |
|---|---|---|---|---|---|
| | B | Robust Std. Err. | Beta | | |
| Constant | 3.993 *** | 0.349 | | 11.46 | ≤0.001 |
| Young children (H1) | 0.564 *** | 0.149 | 0.159 *** | 3.79 | ≤0.001 |
| Employment (H2) | 0.727 *** | 0.150 | 0.220 *** | 4.86 | ≤0.001 |
| Household income (H3) | 0.000 | 0.000 | 0.071 | 1.73 | 0.083 |
| Education (H4) | 0.098 * | 0.039 | 0.098 * | 2.50 | 0.013 |
| Age (H5) | −0.019 *** | 0.004 | −0.197 *** | −4.28 | ≤0.001 |

F (5, 537) = 36.11; *p* ≤ 0.001
R² = 0.252
Root MSE = 1.226

| DV: WTPABS (*n* = 543) | | | | | |
|---|---|---|---|---|---|
| Constant | −3.827 ** | 1.444 | | −2.65 | 0.008 |
| PTP_predicted (H6) | 1.840 *** | 0.290 | 0.278 *** | 6.34 | ≤0.001 |
| PSE (H7) | −0.591 *** | 0.150 | −0.162 *** | −3.94 | ≤0.001 |
| Household income (H8) | 0.000 * | 0.000 | 0.090 * | 2.09 | 0.037 |

F (3, 539) = 34.04; ≤0.001
R² = 0.159
Root MSE = 4.307

Note: * *p* < 0.050, ** *p* < 0.010, *** *p* ≤ 0.001.

### 6.2. Respondents with a Non-Zero WTP

In the first subsample, we now drop the non-users with a WTP of zero, and thus zoom in on the respondents with a non-zero WTP. As can be seen in Table 4, compared to the full sample, several relationships between the demographic factors and PTP are no longer (as) significant. In particular, household income is no longer significant, and education now only impacts respondents' feelings of time pressure at the 10% level.

**Table 4.** PTP for respondents with a non-zero WTP.

| DV: PTP (*n* = 174) | Unstandardized Coefficients | | Standardized Coefficients | t | *p*-Value |
|---|---|---|---|---|---|
| | **B** | **Robust Std. Err.** | **Beta** | | |
| Constant | 4.436 *** | 0.611 | | 7.26 | ≤0.001 |
| Young children (H1) | 0.439 * | 0.201 | 0.165 * | 2.18 | 0.030 |
| Employment (H2) | 0.564 * | 0.264 | 0.175 * | 2.14 | 0.034 |
| Household income (H3) | 0.000 | 0.000 | 0.035 | 0.45 | 0.653 |
| Education (H4) | 0.144 | 0.077 | 0.136 | 1.87 | 0.063 |
| Age (H5) | −0.019 * | 0.008 | −0.195 * | −2.34 | 0.020 |

F (5, 168) = 8.25; $p \leq 0.001$
$R^2$ = 0.197
Root MSE = 1.160

Note: * $p < 0.050$, ** $p < 0.010$, *** $p \leq 0.001$.

As explained, from this step onwards we use a two-stage Heckman procedure to explore our dependent variable WTP (see Table 5). We do not discuss the selection equation(s), as this is not the focus of our paper; moreover, the results are very similar to those of Van Droogenbroeck and Van Hove [33]. The only major difference relates to the presence of household income (a variable that Van Droogenbroeck and Van Hove do not have), which triggers the insignificance of employment. We also tried models without household income and found similar results to Van Droogenbroeck and Van Hove [33] for the selection equations. The most important result—in the outcome equation of Table 5—is that the relationship between PTP and WTP is no longer significant. PSE also has no influence. Conversely, household income is still positively related to WTPABS; however, its impact is negligible. Note that the Wald $Chi^2$ statistic points towards poor quality of the model. Moreover, the likelihood-ratio (LR) test—which shows the comparison of the joint likelihood of an independent probit model for the selection equation and a regression model on the observed WTP data against the Heckman model likelihood—indicates that our choice for a Heckman analysis was in fact not necessary. Indeed, a regression analysis without Heckman correction provides similar results.

**Table 5.** Heckman results for the respondents with a non-zero WTP.

| | Coefficient | Std. Err. | z | *p*-Value |
|---|---|---|---|---|
| **Outcome Equation** (WTPABS) | | | | |
| PTP_predicted | 0.260 | 1.074 | 0.24 | 0.809 |
| PSE | −0.166 | 0.252 | −0.66 | 0.512 |
| Household income | 0.001 * | 0.000 | 2.36 | 0.018 |
| Constant | 6.664 | 7.129 | 0.93 | 0.350 |
| **Selection Equation** (non-zero WTP) | | | | |
| Education | 0.085 | 0.047 | 1.81 | 0.070 |
| Age | −0.020 *** | 0.005 | −4.12 | ≤0.001 |
| Children | 0.415 ** | 0.157 | 2.63 | 0.008 |
| Employment | 0.176 | 0.164 | 1.07 | 0.285 |
| Household income | 0.000 * | 0.000 | 2.52 | 0.012 |
| Constant | −0.517 | 0.394 | −1.31 | 0.189 |
| Athrho | −0.225 | 0.380 | −0.59 | 0.555 |

Wald $Chi^2$ = 5.88 (*p* = 0.118)
Log Likelihood = −781.340

Note: * $p < 0.050$, ** $p < 0.010$, *** $p \leq 0.001$. Censored observations = 369; uncensored observations = 174; LR test of independent Eqns. ($\rho$ = 0): $\chi^2(1)$ = 0.24, Prob > $chi^2$ = 0.6246.

### 6.3. Users

In the final subsample, we narrow down our analysis to the users of click-and-collect services. A first difference with the broader samples is that the only predictors of PTP, in Table 6, are the presence of young children and employment. Household income, education and age have no impact.

**Table 6.** PTP for the users.

| DV: PTP (*n* = 98) | Unstandardized Coefficients | | Standardized Coefficients | t | *p*-Value |
|---|---|---|---|---|---|
| | B | Robust Std. Err. | Beta | | |
| Constant | 3.424 *** | 0.936 | | 3.66 | ≤0.001 |
| Young children (H1) | 0.636 * | 0.260 | 0.258 * | 2.45 | 0.016 |
| Employment (H2) | 0.880 * | 0.364 | 0.250 * | 2.42 | 0.018 |
| Household income (H3) | 0.000 | 0.000 | 0.089 | 0.80 | 0.428 |
| Education (H4) | 0.113 | 0.109 | 0.104 | 1.04 | 0.302 |
| Age (H5) | −0.002 | 0.014 | −0.015 | −0.12 | 0.905 |
| | | | | | |
| $F_{(5, 92)}$ = 4.48; *p* = 0.001 | | | | | |
| $R^2$ = 0.196 | | | | | |
| Root MSE = 1.137 | | | | | |

Note: * $p < 0.050$, ** $p < 0.010$, *** $p \leq 0.001$.

In Table 7 we analyze users' WTP, for which we now have multiple measures, as explained in the Methodology section. Another difference with the broader samples is that the use of predicted values for PTP failed to yield results. A comparison of Tables 3, 4 and 6 show that as the samples become more narrow, the fit of our PTP models deteriorates (as indicated by the F-statistics). This is probably not only due to the drop in the number of observations but also to the low(er) variation in PTP in the users sample (1.52; compared to 1.99 for the full sample and 1.63 for the respondents with a non-zero WTP), which makes it harder to predict their PTP accurately enough. For this sample we therefore simply use the observed values for PTP.

Table 7 shows that the relationships between (observed) PTP and both absolute and relative maximum WTP per order are not significant. However, when we measure WTP as the maximum total service cost per year, PTP *is* a predictor of WTP: consumers' maximum yearly cost increases as they feel more pressed for time (β = 29.188, *p* = 0.013). Remember that for the total service cost per year we used the critical service cost per order (i.e., the fee at which consumers would change their order frequency). Note also that the Wald Chi$^2$ statistic indicates poor quality of the models for WTPABS and WTPREL. The LR test again indicates that the use of Heckman is not strictly necessary (regression analyses without the Heckman correction again yielded similar results).

As to the other hypothesized determinants of WTP, household income has a direct effect on the relative maximum WTP and on the yearly maximum cost, but only at the 10% level. Note that for WTPREL, the impact is negative. When 'income of the respondent' is used, the relationships are not significant. Interestingly, we also find that multichannel users—i.e., consumers who still visit physical supermarkets on a regular basis—have a lower maximum yearly cost than consumers who only buy groceries offline or only when they have forgotten something (β = −228.214, *p* ≤ 0.001). Finally, PSE has an impact on the relative maximum WTP, but only at the 10% level, and the sign is opposed to what we hypothesized (β = 1.442, *p* = 0.055).

**Table 7.** Heckman results for the users.

| | WTPABS (a) | WTPREL (b) | WTPTOT (c) |
|---|---|---|---|
| **Outcome Equation** | | | |
| PTP_observed | 0.368 | −0.526 | 29.188 * |
| | (0.95) | (−0.65) | (2.48) |
| PSE | 0.535 | 1.442 | −13.657 |
| | (1.56) | (1.92) | (−1.34) |
| Household income | 0.000 | −0.001 | 0.019 |
| | (1.23) | (−1.86) | (1.74) |
| Multichannel | −1.107 | −1.033 | −228.214 *** |
| | (−1.17) | (−0.49) | (−8.19) |
| Constant | 3.215 | 14.854 | 129.603 |
| | (0.69) | (1.89) | (0.86) |
| **Selection Equation** (non-zero WTP) | | | |
| Education | 0.111 * | 0.109 | 0.118 * |
| | (1.99) | (1.86) | (1.97) |
| Age | −0.020 ** | −0.023 *** | −0.022 *** |
| | (−3.11) | (−3.49) | (−3.55) |
| Children | 0.568 *** | 0.505 ** | 0.532 *** |
| | (3.54) | (2.72) | (3.21) |
| Employment | 0.165 | 0.188 | 0.145 |
| | (0.87) | (0.95) | (0.72) |
| Household income | 0.000 *** | 0.000 *** | 0.000 *** |
| | (3.23) | (4.26) | (4.22) |
| Constant | −1.590 *** | −1.455 ** | −1.518 *** |
| | (−3.41) | (−3.05) | (−3.18) |
| Athrho | 0.612 | −0.107 | 0.307 |
| | (1.43) | (−0.38) | (0.65) |
| Wald Chi$^2$ | 4.48 | 8.40 | 96.67 |
| P | 0.345 | 0.078 | ≤0.001 |
| Log Likelihood | −488.006 | −564.650 | −818.572 |
| Censored | 445 | 445 | 445 |
| Uncensored | 98 | 98 | 98 |

Note: z statistics between parentheses; * $p < 0.050$, ** $p < 0.010$, *** $p \leq 0.001$.

## 7. Discussion

### 7.1. Level of Willingness to Pay

If we start by looking at the WTU and WTP reported by our respondents, a first observation is that at the time of our data collection (end 2014) the WTU for click-and-collect services was still rather low. In particular, less than one out of five (18.0%) of the non-users were prepared to use such a service at an acceptable price. Although the two numbers are not fully comparable, it is striking that this share is even lower than the 36.2% found by Seitz et al. [12] for German consumers in 2013.

However, the WTP of the non-users who *would* be willing to order groceries online is not marginal. The average reasonable service cost per order is EUR 3.54; the maximum fee is EUR 5.31. Interestingly, for both fees the modus is EUR 5—or only slightly below the current service cost of Collect & Go (and higher than that of its competitors). This suggests that several non-users are not aware of the exact level of the service cost. Retailers might thus benefit from mentioning this cost in their advertising campaigns, all the while emphasizing the advantages of online grocery shopping and the existence of service cost waivers. These waivers work as follows. At specific points in time, the retailer selects a number of products, the purchase of which results in the service cost being waived. For example, if a consumer buys twelve bottles of water of a certain brand, he or she does not have to pay the service cost. Previous research indicates that these service cost waivers are

frequently used by consumers [34]. Another possible explanation for the high modus is that inhibiting factors other than price are stopping non-users from ordering online.

A third observation for the non-users is that those who would be prepared to use an online grocery service at an acceptable price and those who are not, differ mainly in terms of household-level characteristics. In particular, the former live in significantly larger households, and both the share of households with young children and the share of households where both partners work full-time are significantly higher. Where the personal-level demographics are concerned, only age has an impact: non-users with a non-zero WTP are significantly younger. These findings are in line with Van Droogenbroeck and Van Hove [33], who show that personal-level variables affect the ability to adopt, but that consumers' motivations to adopt in fact lie on the household level.

Turning to the users, the average critical price point—i.e., the fee at which they would adjust their behavior—is EUR 8.1; the service cost at which they would completely stop ordering online is on average EUR 10.8. At first sight, supermarkets would thus seem to have some room to increase their service fee. However, they need to take into account that there are large interpersonal differences, and that for 22.4% the critical and maximum fee coincide (for 12 of the 22 respondents for whom the two WTP measures coincide, this happens at EUR 10 or above, but for 10, the point lies below EUR 10—and for some even at EUR 7 or below). Moreover, higher fees would make it less likely that non-users start to use the online channel.

### 7.2. Determinants of PTP

Where our conceptual framework is concerned, overall, the role of demographic characteristics as determinants of time pressure is confirmed. 'Presence of young children' (H2e) and 'employment' (H2b) would seem to be the most important factors, as these variables are strongly significant in all three samples. The importance of young children is in accordance with Morganosky and Cude [112], Raijas [25], and Chintagunta et al. [113], but clashes somewhat with the findings—for Belgium—of Van Droogenbroeck and Van Hove [33], who do not detect an impact of 'presence of young children' on the motivation 'to save time'. Still, they do find that the presence of young children has a positive impact on adoption as such. Van Droogenbroeck and Van Hove also find that the motivation 'to save time' is significantly more important for households where all adults work full-time, which is in line with our result. As for the other demographic characteristics, household income (H2a) is only significant in the full sample and only at the 10% level. This is an indication that it is indeed (as argued in the Literature section) not the best measure of time pressure; although, it is often used in this way in the literature [114]. Finally, we only find partial evidence for the impact of 'age' (H2c) and 'education' (H2d). Both variables impact consumers' PTP in the full sample and in the non-zero WTP sample, but are not significant in the users sample.

### 7.3. Relationship between PTP and WTP

Looking into the main hypothesis of our study—H1 on the relationship between PTP and WTP, our results confirm those of Teller et al. [11] and Seitz et al. [12]. For samples consisting of users and non-users, these authors find, respectively, that time-starved consumers have a higher WTP than 'new technologists', and that the higher consumers' 'need for convenience'—which includes 'saving time'—the higher the probability that they are willing to pay.

For our full sample we effectively find a positive relationship between PTP and the absolute maximum WTP (see Table 3). However, this result would seem to be driven—as may well be the case in Teller et al. [11] and Seitz et al. [12] too—by the presence of a high number of respondents with a WTP of zero. Indeed, the relationship disappears for respondents with a non-zero WTP (in Table 5) and for current users (in Table 7).

Crucially, the result for the users sample—WTPABS does *not* depend on PTP—would seem to present a complication for the popular suggestion in the literature that e-grocers

should price discriminate. However, one could raise the question whether the absolute WTP per order is the best possible measure of users' WTP. It is indeed an open question how users perceive the level of the service cost they have to pay. Do they think about the fee in relative terms, as a percentage of the average amount of their order? Do they factor in the number of times they have to pay it per month or per year? Our two alternative WTP measures can shed light on this question.

For the first alternative, the relative maximum service cost (WTPREL), we again found no impact of PTP. On closer inspection, this is not surprising as there is not only no correlation between PTP and the numerator of the measure (WTPABS; R = −0.025, $p$ = 0.817), but also no correlation between PTP and the denominator (the average amount per order; R = 0.002, $p$ = 0.986). The amount per order also does not differ between users with high levels of PTP (EUR 149.2) and users with low levels (EUR 149.5) (we used a median split with a 5.5 cut-off point). Clearly, the amount per order depends on several other factors, such as household size, income, (brand) preferences, and, as can be seen in Figure A1 in Appendix, order frequency.

It is precisely this order frequency that plays a role in our second alternative measure, the total service cost per year—calculated as the critical price point per order times the number of orders per year. Crucially, for this WTP measure the relationship between PTP and WTP *is* significant (see Table 7). Note also that the Wald Chi$^2$ statistic indicates markedly better quality for WTPTOT than for the other two WTP measures.

The explanation for the significant relationship between PTP and WTPTOT is two-fold. First, the correlation between PTP and order frequency is positive and significant (R = 0.307, $p$ = 0.003). In other words, the higher consumers' level of PTP, the heavier their reliance on the service, and thus the higher their order frequency (see Figure A2 in Appendix A).

Second, while one would, if anything, expect a negative relationship between order frequency and critical price point, the latter is *not* lower for 'heavy users' (see Figure 5). Users who order once a week or more even have the highest critical price point of all (although the difference with the other groups is not significant). This is surprising, as one would expect users who order on a weekly basis to have a relatively low critical price point, because increases in the per-order fee push up the total cost per year quite quickly. An increase of EUR 1 in the service cost increases the yearly cost by at least EUR 52. Conversely, one would expect users who order on a monthly basis to be, ceteris paribus, less sensitive to increases in the service cost, as they only have to pay it twelve times per year.

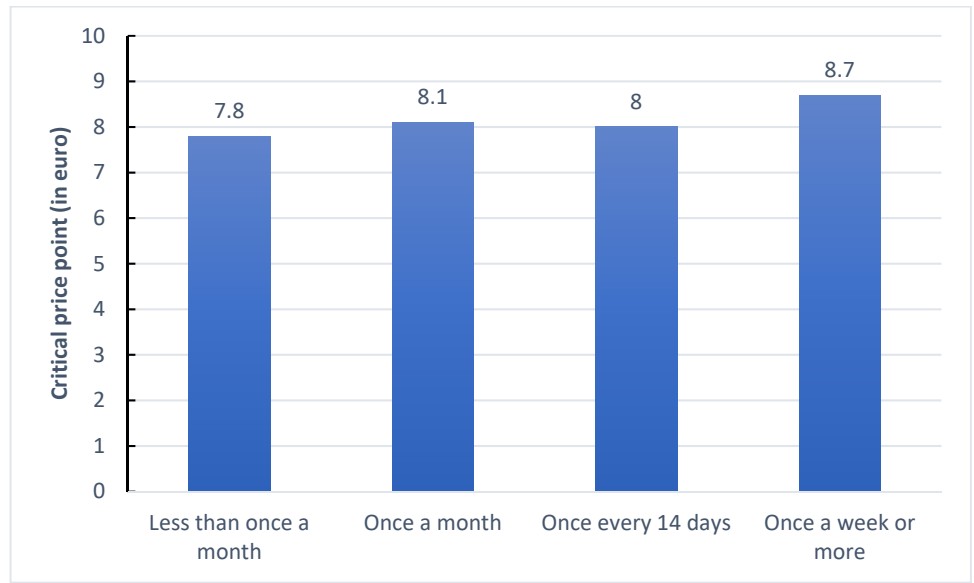

**Figure 5.** Critical price point vs. order frequency.

In addition, users who order once a week or more are probably also the user group who could most easily lower their order frequency without substantially lowering the utility of the service. Users who order, say, five times a month could probably switch to four times without a decrease in the share of their grocery needs that they fulfill via the e-grocery service. Users who now order once a month might find the service less useful at a lower frequency, and might have to visit the physical supermarket more often for top-up purchases. In other words, the fact that the critical price point is not lower for heavy users is, in fact, an indication that they have a higher WTP.

In conclusion, it would seem that—for users—the maximum absolute service cost is not the best measure of WTP; the maximum total service cost over a given period is a better measure. In particular, we find that a higher level of PTP does not translate into a higher WTP per order, but that if the service fees were to increase, time-poor users would maintain their high order frequency for slightly longer than other user groups—in spite of the bigger cost impact.

### 7.4. Other Determinants of WTP

Turning to the other variables that were hypothesized to determine the WTP, we find a negative impact of 'multichannel' (H4) on WTPTOT. In other words, multichannel users have a lower yearly WTP than those who only visit physical supermarkets when they have forgotten something. Unsurprisingly, the latter order more frequently online than multichannel users (see Figure 6). In addition, the critical price point is also higher for the 'pure-online' users (be it that the difference is only significant at the 10% level). Overall, the impact of 'multichannel' is in line with Melis et al. [68], who find that the decision to start ordering groceries online at a particular supermarket chain results in an expansion of the share of wallet assigned to that chain. Moreover, the expansion turns out to be significantly stronger for the time-constrained. In other words, 'pure-online' users seem to be more locked-in on the online channel, and thus have a higher WTP for the service.

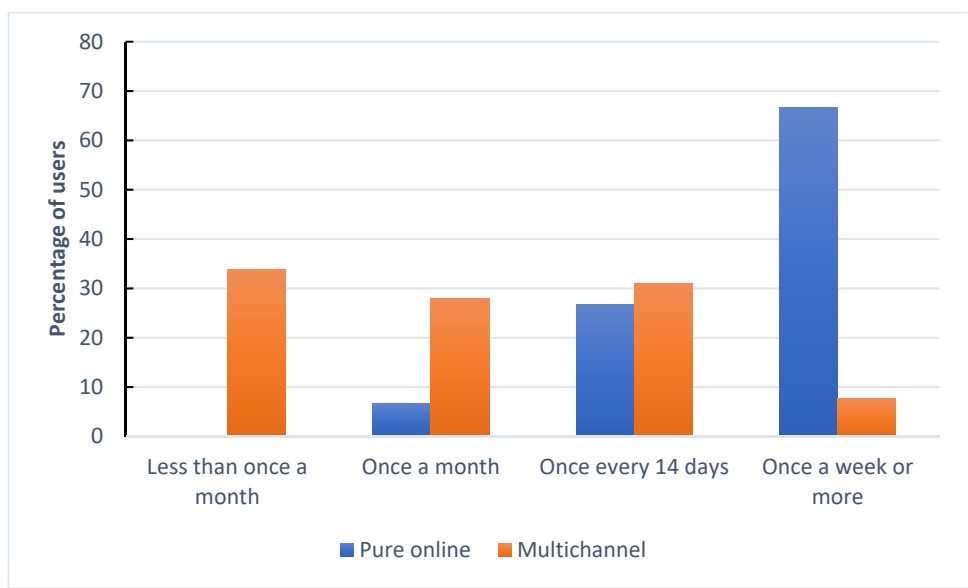

**Figure 6.** Multichannel vs. order frequency.

Where PSE is concerned, for the full sample, we find a direct negative impact on WTP (H3). This finding differs from Teller et al. [11] and Goethals et al. [10], who found no effect of PSE. However, other studies confirm that the dislike of grocery shopping is a motivator to use the online channel [32]. The explanation for our finding would seem to be that the level of PSE is significantly lower for users than for non-users (3.1 vs. 4.0; t(478) = 6.427, $p \leq 0.001$). Consumers who only visit offline supermarkets when they have forgotten

something also have lower levels of PSE than multichannel users (2.5 vs. 3.3; t(95) = −2.877, $p$ = 0.005). In the users sample, the relationship between PSE and WTP is, opposed to what was hypothesized, positive for both WTPABS and WTPREL. For the latter, the relationship is even significant, albeit only at the 10% level. When we measure WTP as the maximum total service cost per year—which is the best measure—the relationship is negative, but not significant.

As for household income (H5), the impact is positive—albeit small—across all three samples (except for WTPREL in the users sample), which is in line with the findings of Seitz et al. [12], but clashes with Teller et al. [11]. However, the impact is only significant at the 5% level in our analysis for the respondents with a non-zero WTP. For the full sample and the users sample the significance level is only 10%.

*7.5. Managerial Implications*

What are the managerial implications of our findings?—apart from the suggestion made earlier that retailers might want to make the level of their service cost better known. As mentioned, the absence, in the users sample, of a significant relationship between PTP and absolute WTP per order would seem to complicate the implementation of price discrimination. However, the PTP of their customers is not something e-grocers could exploit directly anyhow, as they do not observe it. We therefore examine an indicator that e-grocers do observe (and that ties in directly with the suggestions in the literature), namely, whether customers with high levels of PTP use specific time slots.

As the presence of young children and the number of full-time employed adults in the household are positively linked with time pressure, one might expect that time-poor customers pick up their groceries outside work hours or during the weekend. However, although the popularity of time slots outside work hours is effectively higher among respondents with high PTP (50.0%) than among those with low PTP (43.9%), the difference is not significant; $\chi^2$ (1; $n$ = 92; 0.342, $p$ = 0.559). The same is true for pure-online versus multichannel users (51.7% vs. 45.3), and for those who order weekly (54.2%) versus those who order less frequently (48.3%, 38.1%, and 47.4% for, respectively, 'once every 14 days', 'once a month', and 'less than once a month'). We used a dummy variable 'outside work hours' that was coded one, if respondents pick up their groceries on Saturday or during one of the evening slots (17:00–21:00) during the week, and zero otherwise. We also used a stricter version of 'outside work hours', which was coded one, if respondents pick up their groceries during one of the evening slots (17:00–21:00) during the week, and zero otherwise. Again, we find no significant differences.

We also checked what are the most popular time slots: 24.2% of the users in our sample collect their groceries on Friday afternoon (13:00–17:00) or evening (17:00–21:00), or on Saturday. However, again these respondents are not necessarily the ones with a higher PTP: the popularity of the slots does not differ between respondents with low PTP (24.4%) and those with high PTP (23.5%); $\chi^2$ (1; $n$ = 92; 0.009, $p$ = 0.923).

In short, there is no clear pattern between PTP and the use of specific time slots. Together with the limited difference in the maximum service cost per order between 'high PTP' and 'low PTP' users, this indicates that the application of price discrimination as suggested in the literature is unlikely to substantially improve e-grocers' profitability. Note that the suggestion to charge higher fees for popular and/or rush-hour time slots boils down to second-degree price discrimination, also known as versioning. The basic idea is that the e-grocer creates two versions of the service: a high-quality option at a higher price, and a lower-quality option at a lower price—the difference in quality being the convenience of the time slots. Users would then reveal their WTP through the version they select—those with a high WTP would make use of the more expensive time slots, and *vice versa*—and the e-grocer would extract more consumer surplus compared to the setting with uniform fees for all time slots.

However, if the case that we study can be generalized, the effectiveness of such an approach is probably limited. For one, the envisaged segmentation is unlikely to happen

because we find that, in a setting without price discrimination, several 'high PTP' users make use of less popular and/or off-peak time slots. In other words, they do not perceive these slots as inconvenient and 'low-quality'—a necessary condition for price discrimination to work. Concretely, in our sample only 50.0% of the 'high PTP' users collect their groceries during an 'outside work hours' time slot. Of these users only 46.2% make use of the popular Friday/Saturday slots. Or, looking at it from a different angle, of the users of the 'outside work hours' slots only 59.1% are 'high PTP'. Similarly, of the users of the Friday/Saturday slots only 54.6% are 'high PTP'.

Second, the absence of a significant correlation between users' PTP and the maximum service cost per order they are willing to pay (see Figure 4) implies that supermarkets could not increase their service cost for the popular slots by a substantial amount without triggering a reaction from quite a few 'high PTP' users, who would switch to the cheaper time slots or lower their order frequency. Users with a fixed time slot that coincides with one of the affected slots might even stop using the service altogether if they have a low WTPABS.

On a final note, failing second-degree price discrimination, e-grocers could in principle try to exploit our finding that there is a significant positive relationship between PTP and WTPTOT by charging a service cost that increases with usage. Colruyt could, for example, charge the current EUR 5.50 for the first 12 orders within a year, and charge a higher (also fixed) fee for any additional orders. This would amount to third-degree price discrimination, whereby 'heavy users' (those who order 'once every 14 days' and 'once a week or more') would pay more than the others. However, such a strategy is unlikely to be well-received, as this is not the way loyal customers expect to be treated, on the contrary. Moreover, here too, the room for a fee increase would seem limited (see Figure 5).

## 8. Conclusions

Research into consumers' WTP for online grocery services is relevant in view of the fierce competition in the (online) grocery landscape and the fact that e-grocers struggle with their costs. More in particular, a popular suggestion in the literature is that supermarkets should price discriminate across time slots and days in order to steer demand and improve profitability. However, only a few studies have actually investigated consumers' WTP, and most of the focus has been on home delivery services. The present paper is, to the best of our knowledge, the first to investigate how much consumers are willing to pay for click-and-collect services—a model dominant in countries with high store density, such as Belgium and France. In particular, we examine whether time-poor consumers are willing to pay higher fees. Data were collected among 572 customers of two Belgian supermarkets—both users (105) and non-users (467)—and analyzed by means of systems of regression analysis.

The main conclusions are as follows. One, at the time of our data collection the WTU for click-and-collect services was still rather low, in that only 18% of the non-users were prepared to use such a service at an acceptable price. Second, those who would be prepared to do so and those who are not differ mainly in terms of household characteristics, namely, presence of children, relative number of full-time employed partners, and household size.

Third, where our main hypothesis is concerned, for the full sample—which includes both users and non-users, we effectively find a positive relationship between PTP and the maximum absolute WTP. Interestingly, this correlation disappears for respondents with a non-zero WTP and for current users. On closer scrutiny, for the users, we do find a positive relationship between PTP and WTP. However, crucially, this does not show up in the form of a significantly higher WTP per order, but as a willingness to pay a higher cost per year in the face of increasing fees. Indeed, we find that, if the service fees were to increase, time-poor users would maintain their higher order frequency for as long as other users (and even slightly longer).

Together with the absence of a clear link between PTP and the use of specific time slots, our findings imply that the effectiveness of price discrimination as suggested in the literature is doubtful—at least in the case that we study. For one, the absence of a significant correlation between users' PTP and the maximum service cost per order they are willing to pay implies that e-grocers cannot increase the service cost for popular time slots by a substantial amount without driving away quite a few 'high PTP' users. In addition, with the current price setting several 'high PTP' users pick up their groceries during less popular and/or off-peak time slots, which means they would not be affected by the higher fees. In short, the capturing of consumer surplus would probably be limited.

Fourth, where the other variables in our conceptual framework are concerned, the 'presence of young children' and 'employment' are found to be the most important demographic determinants of time pressure. We also find evidence for the impact of 'household income', 'multichannel', and 'perceived in-store shopping enjoyment' on consumers' WTP for click-and-collect services.

Overall, our paper makes significant contributions to the literature on the last-mile issue—the delivery to the customer's home. That is, while several studies evaluate the financial and environmental performance of different last-mile distribution concepts, our study is the first to investigate the issue from the customers' point of view by exploring WTP in the e-grocery sector. To date, the literature on consumers' WTP for e-grocery services is scant. The majority of the existing studies focus on home delivery services and they only question consumers whether they would be willing to pay or not. Our study extends the theoretical knowledge by studying the level of, and variation in, consumers' WTP for click-and-collect grocery services. In addition, it also explores the opportunities for e-grocers to price discriminate.

In terms of methodological contributions to the literature, our results clearly demonstrate the need for additional academic research on the topic, as the issue proves to be considerably more complex than previously thought. Apart from the suggestions below, future research might in particular look into how consumers perceive the service cost they have to pay. That is, do they think about it in absolute or in relative terms? If it is the latter, what do they use as their point of reference?

Obviously, our study has a number of limitations. First, the generalizability of our results is not guaranteed. As we restricted ourselves to Brussels and Flanders, our sample is not representative for the Belgian population. Moreover, where the users sample is concerned, the large majority are customers of Collect & Go. Further research could benefit from a larger sample of users from multiple services. Researchers could then also control for the type of grocery store, such as soft discounters versus traditional supermarkets. On a higher level, our data are limited to the specific case of Belgium. As a consequence, our findings are not necessarily true for other countries, in which, for example, the click-and-collect model is less dominant. More research, in other countries and other cultures, is needed to validate the results.

Second, we only questioned respondents about the fee at which they would start adjusting their behavior and about the fee at which they would completely stop using the service. We have no information about respondents' precise reaction path between their critical price point and their maximum WTP. As a result, our preferred WTP measure—the maximum total cost per year—may be an overestimation for some respondents and an underestimation for others. Further research could inquire not only at what fee level consumers would change their order frequency but also by *how much* they would lower it.

Third, we did not examine if and how users would switch time slots when confronted with price discrimination. Experimental research could confront respondents with an overview of days and time slots for pick-up and then ask them, for a pre-determined set of fees, during which slot they would collect their groceries. Such an approach would allow researchers to further investigate the possibilities for price discrimination by exploring the link between PTP and the WTP *for particular time slots*.

**Author Contributions:** Conceptualization, E.V.D. and L.V.H.; methodology, E.V.D. and L.V.H.; formal analysis, E.V.D.; investigation, E.V.D.; writing—original draft preparation, E.V.D. and L.V.H.; writing—review and editing, E.V.D. and L.V.H.; supervision, L.V.H. All authors have read and agreed to the published version of the manuscript.

**Funding:** This research received no external funding.

**Institutional Review Board Statement:** Not applicable.

**Informed Consent Statement:** Not applicable.

**Data Availability Statement:** The data presented in this study are available on request from the corresponding author.

**Conflicts of Interest:** The authors declare no conflict of interest.

## Appendix A

*Appendix A.1. Multi-Item Scales*

PTP—adapted from Van Kenhove and De Wulf [64] and Verhoef and Langerak [67]:
I usually find myself pressed for time;
I am often in a hurry;
Usually there is so much to do that I wish I had more time;
For different reasons, I do not have enough time for grocery retail shopping.
PSE—adapted from Spangenberg et al. [106]:
I find shopping for groceries (in a physical store/supermarket) rather

| | |
|---|---|
| Dull | Exciting |
| Not Fun | Fun |
| Not amusing | Amusing |
| Not enjoyable | Enjoyable |

*Appendix A.2. Users' Choice of Time Slots*

**Table A1.** Fixed moment to collect the groceries.

| | Fixed Time Slot? | | | | |
|---|---|---|---|---|---|
| **Fixed Day?** | **No Fixed Time Slot** | **08:00–13:00** | **13:00–17:00** | **17:00–21:00** | **Total** |
| No fixed day | 19 | 6 | 6 | 16 | 47 |
| Monday | 1 | 3 | 1 | 5 | 10 |
| Tuesday | 0 | 0 | 0 | 2 | 2 |
| Wednesday | 0 | 3 | 1 | 2 | 6 |
| Thursday | 2 | 0 | 0 | 2 | 4 |
| Friday | 1 | 5 | 4 | 7 | 17 |
| Saturday | 0 | 8 | 5 | 0 | 13 |
| Total | 23 | 25 | 17 | 34 | 99 |

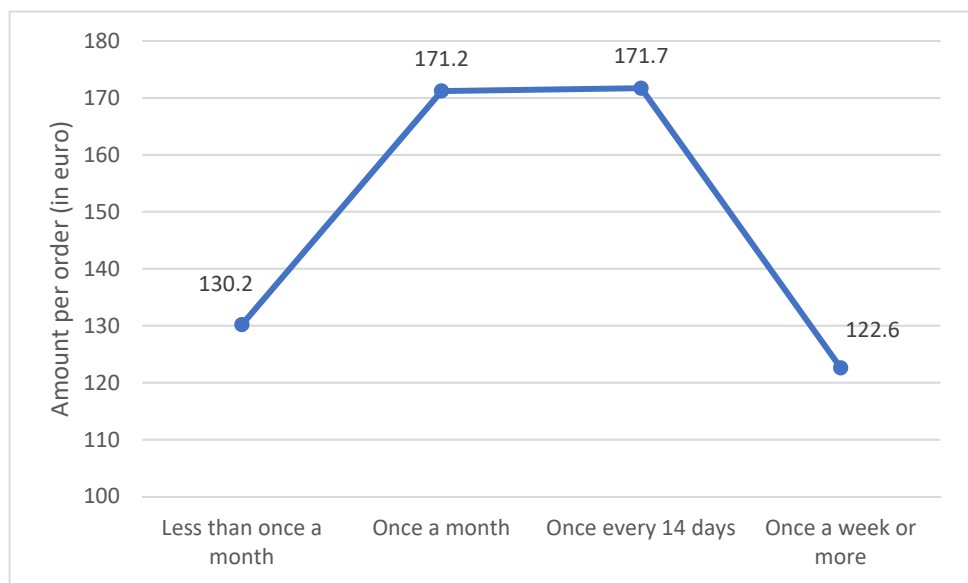

**Figure A1.** Average amount per order vs. order frequency.

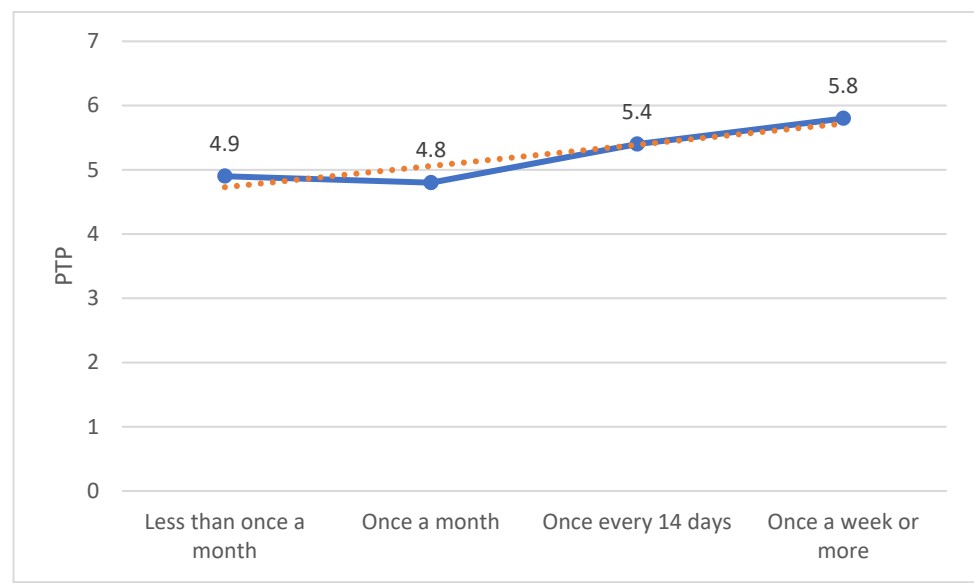

**Figure A2.** PTP vs. order frequency.

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
