# Peer review of "Are the Time-Poor Willing to Pay More for Online Grocery Services? When ‘No’ Means ‘Yes’"

_jtaer, doi:10.3390/jtaer17010013_

Round 1
Reviewer 1 Report
Generally, the article is well written, the authors may consider the following comments:
- May consider reducing the size of the article from 50 pages to at least 30 pages. The paper looks like a thesis.
- Line 5, should it be 1 or a, please make this clear.
- Line 32, please explain the meaning of EU27, the reader would need to understand the meaning of the abbreviation.
- Line 63, may be named 2.1 The logistics
- Line 123, may be named 2.2 How convenient is e-grocery shopping
- Line 173, may be named 2.3 Willingness to pay for e-grocery....
- For the Conceptual framework and Hypotheses: Make the hypotheses clear and separate from the rest of the text.
- Methodology: Consider using 4.1; 4.2;
- Descriptive results: Consider using 5.1; 5.2; 5.3
- Discussion: Consider using 7.1; 7.2; 7.3; 7.4; 7.5
Reviewer 2 Report
The authors of the paper Are the Time-poor Willing to Pay More for Online Grocery Services? When 'No' Means 'Yes' presents a relevant topic, namely "investigating the willingness to pay (WTP) of consumers for food services", especially in the current pandemic context, but especially due to the fact that the resource time becomes increasingly valuable for those involved in multiple activities.
Bibliographic sources, citations and concepts are used appropriately by the authors of the research. Specifically, sources such as those of the authors „Teller et al. (2006) find that time-limited consumers have a higher WTP than “new technology”.
The research methodology is adequately presented, the authors of the research using relevant data, appropriate tools such as the survey accompanied by the questionnaire. Furthermore, independent variables are highlighted in the questionnaires, such as “education is a categorical variable”.
The results of the research are presented by the authors of the research on the basis of surveys and the interpretation of questionnaires, and as a result of these "samples of users and non-users, time-consuming consumers have a higher WTP than" new technologies "and that" the greater the "need for convenience" of consumers - which includes "saving time", the more likely they are to be willing to pay ". However, we suggest the authors of the research to highlight the main scientific results as a personal contribution to the scientific literature, given that the results are adequately presented but very much oriented on the application side.
The findings are presented by the authors of the study, who emphasized "how much consumers are willing to pay for click-and-collect services - a dominant pattern in high-density store countries such as Belgium and France". At the same time, the authors adequately present the limitations of the study, as well as future research. Moreover, as we mentioned in the results chapter, we appreciate that the personal scientific results that contribute to the scientific literature should be highlighted.
We congratulate the research team, we suggest the revision of the paper according to the above mentioned, and after the revision we propose for acceptance the paper.
